# Low and high beta rhythms have different motor cortical sources and distinct roles in movement control and spatiotemporal attention

**Simon Nougaret**[1]*, **Laura López-Galdo**[1], **Emile Caytan**[1¤a], **Julien Poitreau**[1¤b], **Frédéric V. Barthélemy**[2,1], **Bjørg Elisabeth Kilavik**[1]*

**1** Institut de Neurosciences de la Timone (INT), UMR 7289, Aix-Marseille Université, CNRS, Marseille, France, **2** Institute of Neuroscience and Medicine (INM-6), Jülich Research Centre, Jülich, Germany

¤a Current address: University of Crete, Faculty of Medicine & Foundation for Research and Technology Hellas, Institute of Applied and Computational Mathematics, Heraklion, Greece
¤b Current address: Laboratoire de Neurosciences Cognitives (LNC), UMR 7291, CNRS, Aix-Marseille Université, Marseille, France
* Simon.Nougaret@univ-amu.fr (SN); Bjorg.Kilavik@univ-amu.fr (BEK)

**Data Availability Statement:** Data for the reproduction of main and supplementary figures

## Abstract

Low and high beta frequency rhythms were observed in the motor cortex, but their respective sources and behavioral correlates remain unknown. We studied local field potentials (LFPs) during pre-cued reaching behavior in macaques. They contained a low beta band (<20 Hz) dominant in primary motor cortex and a high beta band (>20 Hz) dominant in dorsal premotor cortex (PMd). Low beta correlated positively with reaction time (RT) from visual cue onset and negatively with uninstructed hand postural micro-movements throughout the trial. High beta reflected temporal task prediction, with selective modulations before and during cues, which were enhanced in moments of increased focal attention when the gaze was on the work area. This double-dissociation in sources and behavioral correlates of motor cortical low and high beta, with respect to both task-instructed and spontaneous behavior, reconciles the largely disparate roles proposed for the beta rhythm, by suggesting band-specific roles in both movement control and spatiotemporal attention.

## Introduction

A link between the beta rhythm in human sensorimotor cortex and voluntary movements was established 75 years ago [1]. Yet, the functional role of sensorimotor beta remains elusive. Beta was associated with many aspects of motor behavior, ranging from motor cortical idling or postural maintenance [2–11] to sensorimotor integration or temporal predictions [12–22]. As beta rhythms were observed in many cortical and subcortical regions and in many different behavioral contexts, they might serve multiple roles [23–26].

Most studies have treated the broader beta frequency range (approximately 13 to 35 Hz) as one common motor cortical rhythm. This has hindered the association of specific beta

and custom code used to compute the aperiodic estimation (Fig 3) are available in S1 Data, S2 Data and S1 Script. The toolboxes and functions of Matlab and Python used for other analysis are detailed in the Materials and Methods section. A repository with preprocessed datasets including all LFP sites and the related behavior is available at https://doi.org/10.12751/g-node.wvz4wd.

**Funding:** EU Horizon 2020 Marie Skłodowska-Curie Actions grant In2PrimateBrains #956669 (BEK). https://cordis.europa.eu/project/id/956669/fr FLAG-ERA grant PrimCorNet ANR-19-HBPR-0005 (BEK). https://anr.fr/Projet-ANR-19-HBPR-0005 None of the funders played any role in the study design, data collection and analysis, decision to publish, or preparation of the manuscript.

**Competing interests:** The authors have declared that no competing interests exist.

**Abbreviations:** BIC, Bayesian index criterion; DAQ, data acquisition; ECoG, electrocorticography; EMG, electromyogram; ICMS, intra-cortical electrical microstimulation; LED, light-emitting diode; LFP, local field potential; LM, linear model; MUA, multi-unit activity; NSP, neural signal processor; PMd, premotor cortex; RT, reaction time; SC, spatial cue; SEL, selection cue; t-SNE, t-distributed stochastic neighbor embedding.

frequencies to specific aspects of sensorimotor behavior and prohibited building a unified theory regarding the role(s) of sensorimotor beta. A few studies divided this broad band into low beta (below 20 Hz) and high beta (above 20 Hz). In a published dataset [27,28], we observed concurrent and distinct low and high beta bands during visuomotor behavior in macaque motor cortical local field potentials (LFP). However, the low and high bands modulated similarly in power and peak frequency in that behavioral task. Also Stoll and colleagues [29] observed 2 distinct low and high beta bands in macaque frontal cortical electrocorticography (ECoG), in a trial-and-error task comprising search and repetition phases. They found only the high band to be systematically sensitive to attentional effort and cognitive control. Chandrasekaran and colleagues [30] correlated behavioral reaction time (RT) with dorsal premotor cortex (PMd) beta power, in an RT-task. Their data contained a single band peaking at 25 Hz, which shifted slightly towards higher peak frequency in the pre-stimulus period for shorter RT, resulting in positive correlations below 20 Hz and negative correlations above 20 Hz. In comparison, Zhang and colleagues [31] found positive correlations with RT for sensorimotor pre-stimulus alpha/beta power covering 8 to 33 Hz in an RT task.

These studies remain far from conclusive in determining potentially distinct correlations between behavior and motor cortical low and high beta bands. We therefore designed a new visuomotor behavioral task to maximize at the same time spatiotemporal attention and the required motor control, with the aim of disentangling which of the different task variables, and task-instructed and uninstructed (spontaneous) behavioral factors [32,33] affect the 2 beta bands. We hypothesized that low beta might be related to dynamic postural control and movement preparation. This band was shown to be more affected in human Parkinson's disease patients than high beta [34,35], and more attenuated by stimulation in the subthalamic nucleus [36] and levodopa administration [37], 2 therapies that improve motor control. Furthermore, low beta is attenuated after kinematic errors [38]. In contrast, motor cortical high beta might be more closely associated with attention, working memory, decision-making or reward [29,38–40], and originate anterior to low beta [41]. Consistent with these predictions, we found low beta to be dominant in primary motor cortex (M1) and correlate positively with behavioral RT in 2 macaques. Low beta also correlated negatively with spontaneous hand postural micro-movements that were frequent during the maintenance of stable central hold during delays. High beta, on the other hand, was dominant in PMd, and was unrelated to RT and hand postural micro-movements. Instead, it modulated selectively during anticipation and processing of visual cues. This modulation was enhanced by focal overt attention, when the animal oriented the gaze towards the work area. We conclude that motor cortex contains multiple, independent beta rhythms operating simultaneously, with band-specific behavioral correlates.

## Results

We studied LFP low and high beta band rhythms recorded in the motor cortex (M1 and PMd) of 2 macaque monkeys engaged in a complex visuomotor reaching task (Fig 1A). We determined the motor cortical regions in which each band dominated, and we quantified their relationship to task conditions and performance, and to spontaneous hand and eye movements.

### Behavioral task performance

Two macaque monkeys performed a delayed match to sample task with fixed cue order and a GO signal, requiring arm reaching responses in one of 4 (diagonal) directions from a common center position. The visuomotor task was complex, requiring the animal to select the valid spatial cue (SC; one out of three sequentially presented SCs; the other 2 were distractors) that

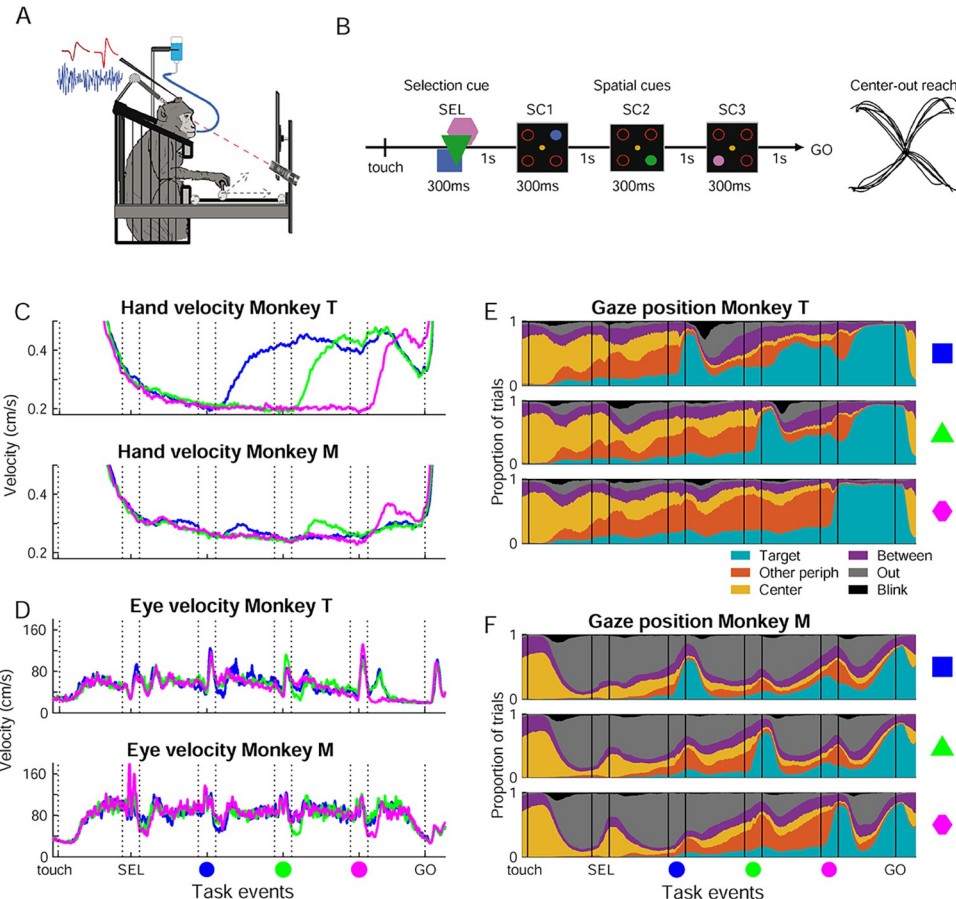

**Fig 1. Experimental setup and task, spontaneous hand and eye movements.** (A) The monkeys were seated in a primate chair and performed center-out arm reaching responses with a manipulandum in the horizontal plane, with the visual scene displayed on a vertical monitor. Eye position was recorded using an infrared camera. (B) The monkeys performed a visuomotor delayed match to sample task with fixed cue order and a GO signal. The trial started when the monkey moved the hand cursor to the central fixation spot (touch). Next, a selection cue (SEL) indicated the color to attend in that trial (sample). Thereafter, 3 spatial cues (SC) were presented in sequence in fixed order (blue SC1 –green SC2 –pink SC3), each in one of the 4 possible peripheral target positions. A directionally non-informative GO signal indicated to the monkey to initiate the center-out reaching movement to the memorized valid (match) target location. Each delay lasted 1 s and each visual cue lasted 300 ms. (C) Average hand velocity across all trials in all behavioral sessions for each monkey, zoomed in to the micro-movements performed during the trial between central touch and the GO signal. In this and subsequent figures, the blue, green, and pink lines reflect data split according to the color condition and vertical lines reflect onset/offset of task events. (D) Average eye velocity for each monkey across all trials in all behavioral sessions with eye movement recordings. Same conventions as in C. (E) Gaze position for monkey T, across all trials in all behavioral sessions with eye movement recordings, in blue (upper), green (middle), and pink (bottom) color conditions. Each plot show the proportion of trials with eye gaze on the Target SC (cyan), on one of the other peripheral target outlines (orange), on the central fixation spot (yellow), between different visual items but within the work area (purple), outside the work area (i.e., outside the monitor; gray), or eyeblinks (black). (F) Gaze position for Monkey M. Same conventions as in E. The monkey in A was drawn by hand, inspired by an image by Wirestock on Freepik.com. Source data are available in S1 Data. SC, spatial cue; SEL, selection cue.

matched in color with a preceding color selection cue (SEL; Fig 1B). The animal then had to prepare a center-out arm reaching movement to the memorized matching SC position, to be executed after a GO signal. Throughout the sequence of presentation of the different visual cues, the animal had to maintain central hand position with the manipulandum within a very small zone, but was free to explore the visual scene with the eyes.

We analyzed 59 sessions in monkey T and 39 sessions in monkey M. Of all initiated trials, central hand position maintenance was lost before the GO signal in about 40% (S1 Table), reflecting the difficulty of initial stabilization and maintenance of the hand manipulandum within the 0.6 cm diameter fixation zone. Several types of errors were also made in trials not aborted before the GO signal (GO trials). Of these, directional errors towards a distractor (distractor errors) were in majority (about 20% of GO trials, S1 Table). These occurred less frequently when the third and last SC (pink) was valid ($p < 0.01$ for both) and somewhat less for movements towards the body (i.e., lower visual field; $p < 0.01$ for monkey T, $p = 0.036$ for monkey M).

Even if our task entailed an explicit GO signal, dissociated in time from the informative cues, trial-by-trial fluctuations in RT might reflect the level of motor readiness at the time of the GO signal, influenced by movement preparation processes and/or overall level of alertness or fatigue. We quantified the variability in RT across sessions, color conditions, and movement directions. All 3 factors influenced RT, in a similar manner in the 2 animals ($p < 0.01$; S1 Table). First, there was a main effect of the session, with different average RT in different sessions, but with no trend of increasing or decreasing RT from early to late sessions. Furthermore, RT was shorter in the pink color condition and for movements towards compared to away from the body.

Finally, the RT increased with time-on-task [29,42], being positively correlated with the trial number within the session ($p = 0.022$ for Monkey T, $P < 0.01$ for Monkey M).

## Spontaneous hand and eye movements

Both animals made spontaneous (uninstructed) movements during the behavioral trial. This included hand micro-movements during the maintenance of the hand cursor within the central fixation spot (Fig 1C), and gaze shifts to and from the work area (computer monitor), and between the items of the visual scene (Fig 1D–1F). Although uninstructed, these hand and eye movements were aligned to task events and were remarkably similar in the 2 animals.

We used velocity to quantify the hand micro-movements. The trial-averaged hand velocity decreased as the hand stabilized inside the central fixation at trial start and was minimal at the onset of the valid SC. After valid SC presentation, hand velocity increased, and differed significantly for the 3 conditions (Figs 1C and S1A–S1C). These micro-movements did not reflect a drift of the hand position in the (diagonal) direction of the upcoming center-out reaching movement (S1D Fig), unlike the spatial attention effects described for eye fixational microsaccades [43]. Instead, the hand prevalently drifted along one or the other main axes defined by the 2D manipulandum, having lower frictional resistance than for diagonal movements involving both axes. Control electromyographic (EMG) recordings from one of the proximal muscles involved in the task (deltoid) revealed increased muscular tone during the preparatory period following the valid SC onset (S1E Fig). This increase was similar for preparation of movements towards and away from the body, contrasting with the strong directional selectivity of this muscle during the center-out reaching execution. Thus, the hand micro-movements during preparation were probably related to increased muscle tone of arm muscles involved in the subsequent reaching movement.

The monkeys frequently made eye movements to explore the items of the visual scene or to shift the gaze In/Out of the work area (Fig 1D–F). The gaze was often directed Out, possibly reflecting moments with less focal attention on the task. Monkey M spent more time gazing Out, but less before and during the valid SC and as the GO signal approached. Finally, both monkeys restricted eyeblinks to the delays, thus under tight temporal attention control, as also shown in humans engaged in demanding working memory tasks [44].

Since both hand micro-movements and gaze position (In/Out) modulated in distinct manners for each color condition, their temporal profile in single trials could be used to decode the condition (S2 Fig).

## Concurrent low beta dominant in M1 and high beta dominant in PMd

LFP activity was first analyzed in each recording site to determine any link between the anatomical location of the site and the frequency specificity (high versus low beta band). LFP activity from 110 recording sites (59 sessions) in monkey T and 60 sites (39 sessions) in monkey M was considered. Spectrograms for 1 session with 3 simultaneously recorded sites (Fig 2A–2C) showed a high beta band (>20 Hz) predominant in the most anterior site (PMd; site 1), and a low beta band (<20 Hz) predominant in the most posterior site (M1; site 3). Both bands were distinguishable in the intermediate site (site 2). Even if the trial-averaged spectrograms showed increased beta band power across long periods of the task, single-trial LFPs showed bursts of high beta in PMd (Fig 2D) and low beta in M1 (Fig 2F) of variable durations and timing across trials, as already described [13,28,45,46]. Grand average spectrograms across all trials and LFP sites for each monkey (Fig 2E) clearly contained a low and a high beta band in both animals.

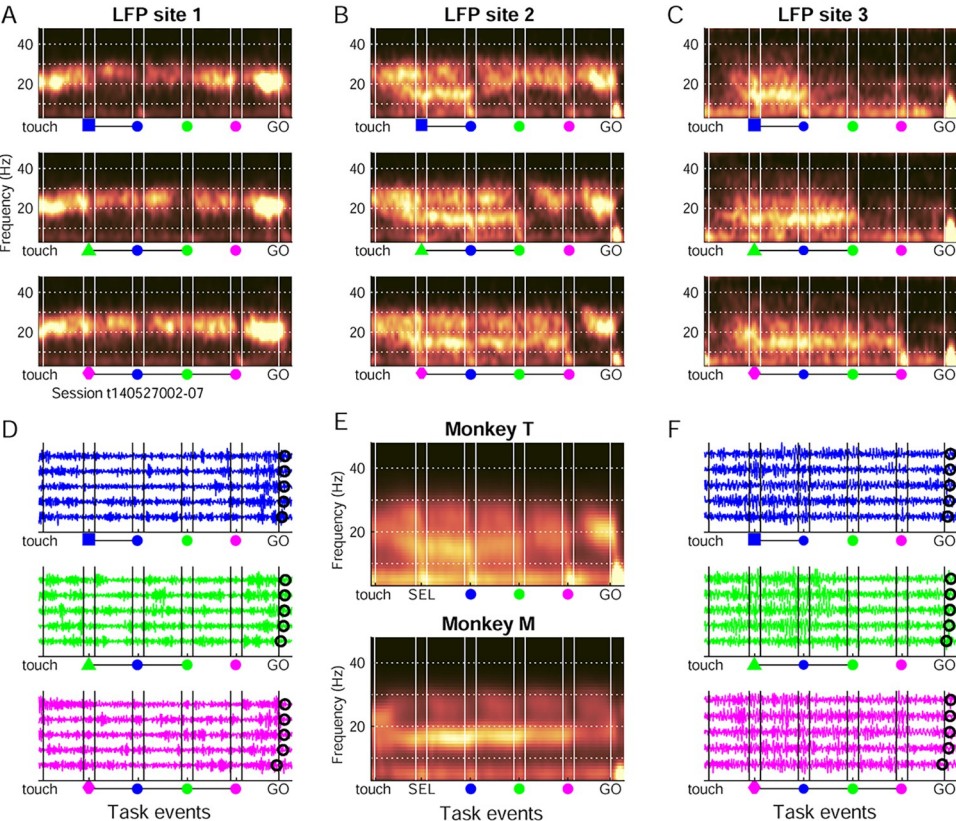

**Fig 2. Example LFP sites and grand average spectrograms.** (A–C) Spectrograms of 3 simultaneously recorded LFP sites from monkey T, including all correct trials in 1 session, separated for the blue (top), green (middle), and pink (bottom) color conditions. The locations of the 3 example LFP sites are marked with stars in Fig 3C, with site 1 more anterior and site 3 more posterior. Frequency is on the vertical axis and task events are indicated along the horizontal axis (vertical white lines). Warmer colors indicate increased power (a.u.). (D) Single trial examples of LFPs filtered broadly around the beta frequency range (8–45 Hz), for LFP site 1. Five trials per color condition are shown. (E) Grand average spectrograms for each monkey, including normalized individual trials for all LFP sites in each monkey. (F) As in D, but for LFP site 3. Source data are available in S1 Data. LFP, local field potential.

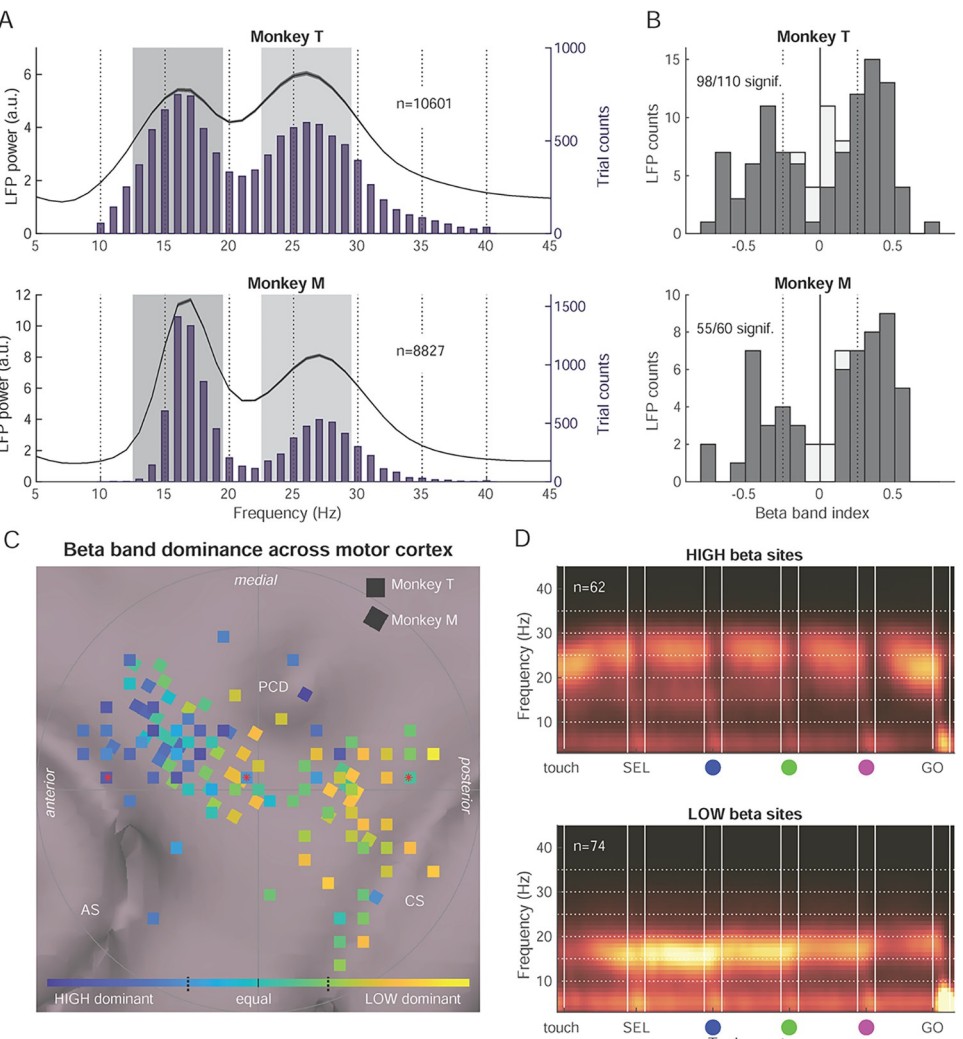

**Fig 3. Concurrent low and high beta band rhythms in the motor cortex.** (A) Average normalized power spectra in the pre-SC1 period across all trials (*n*) for all sites in each monkey, after removing the aperiodic signal component. The curves reflect the mean power ±SEM across LFP sites. Overlain are distributions of single-trial peak frequency (frequency with maximal power) between 10 and 40 Hz in the same task period, in the periodic-only signal component. (B) Distribution of the beta band dominance index for all LFP sites (*n*) for each monkey, based on the periodic-only signal component in the pre-SC1 period. Positive indices reflect low band (13–19 Hz) dominance and negative indices reflect high band (23–29 Hz) dominance. Light gray bars include all sites, and darker gray bars only sites with significantly different power in the low and high beta frequency ranges (paired *t* test, *p* < 0.05). (C) Beta band dominance indices plotted as a function of LFP site coordinates on the cortical surface. The indices for both monkeys are plotted on top of the MRI-based cortical surface reconstruction of monkey T (anterior towards the left and medial towards the top). Blue sites reflect high band dominance, and yellow sites low band dominance. The 3 sites from anterior to posterior marked with red stars reflect the example sites shown in Fig 2. CS central sulcus; AS arcuate sulcus; PCD precentral dimple. (D) Grand average spectrogram for both monkeys combined, for sites (*n*) with strong high beta dominance (indices <-0.25; top), or strong low beta dominance (indices >+0.25; bottom). The weakly dominant sites (indices within +/− 0.25) are not shown. Vertical dotted lines in B and on the color bar in C indicate the index thresholds for the sites included in D. Source data are available in S1 Data. LFP, local field potential.

The grand average, periodic-only power spectra confirmed a low band peaking at about 16 Hz, and a high band at about 26 Hz (Fig 3A). We computed a *beta band dominance index*, after first removing the aperiodic signal component (see Methods), in the delay before SC1 (blue). A small majority of sites had higher power for the low beta band (64/110 sites in

monkey T and 38/60 in monkey M; Fig 3B), the remaining had higher power in the high beta band. We then superimposed the beta band dominance index on the cortical surface reconstruction within the recording chamber. This revealed a gradient with high beta dominant in the anterior recording sites (PMd), and low beta dominant in the posterior (M1) and intermediate sites (Fig 3C). The band dominance index correlated significantly with antero-posterior site coordinate within the recording chamber for each animal ($p < 0.01$; Spearman's rank order correlation). To confirm local origin of these LFP beta rhythms, we analyzed phase-locking of neurons to the locally dominant beta band, for neurons and LFPs recorded on the same linear array. For the high band dominant sites, 46.0% of neurons (27/66 neurons in monkey T and 45/91 in monkey M) were significantly phase-locked to high beta phase. For the low band dominant sites, 12.3% of neurons (22/269 neurons in monkey T and 38/218 in monkey M) were significantly phase-locked to low beta phase. To summarize, 2 beta bands coexist in the motor cortex, with low beta dominant in posterior sites (M1) and high beta dominant in anterior sites (PMd).

## Distinct amplitude profiles of high and low beta

The next analysis focused on the amplitude profiles of high and low beta rhythms along the task. We investigated how LFP low and high beta during individual correct and error trials reflected the sequentially presented visual stimuli and the behavioral choices of the monkeys.

Low and high beta peak frequencies (Fig 3A) and beta band dominance across the cortical surface (Fig 3C) were similar in the 2 animals. Since the task performance (S1 Table) and spontaneous hand and eye movements (Fig 1C–1F) were also similar in the 2 monkeys, from hereon we collapsed the data for the 2 animals. We combined all individual trials from all LFP sites with the same beta band dominance, such that each LFP site was assigned to either contribute to the low or the high band. The grand average spectrograms, for both monkeys combined, with either strong high (indices <-0.25) or low beta dominance (indices >+0.25), are shown in Fig 3D. For further analysis, the normalized, single-trial instantaneous beta amplitude was calculated (Hilbert transform). The trial-averaged amplitude including all sites from both monkeys differed distinctly for low and high beta (Figs 4A, S4A, and S4C). The amplitude of the high band was strong right from the trial start and throughout most trial epochs, only dropping temporarily around each SC. Thus, the high band reflected the "rhythmic" nature of the sequence of task events. In comparison, the amplitude of the low band increased gradually after trial start and was maximal between SEL and valid SC. It then dropped selectively after the valid cue and remained lower through the rest of the trial. Both bands had minimal amplitude after GO. Each band had a distinct temporal profile and, importantly, they were not correlated with each other at the single trial level (S5 Fig). Each of the 2 temporal profiles of average beta amplitude were accompanied by quasi inverse temporal profiles of grand average local multi-unit activity (MUA) amplitude (S6 Fig). MUA amplitude in high beta dominant sites peaked during valid SC presentation, while MUA amplitude in low beta dominant sites was minimal between SEL and valid SC and elevated during movement preparation. The distinctly different profiles of average MUA amplitude obtained when splitting sites according to beta band dominance add support to an at least partly local origin of the 2 beta bands.

A dimensionality reduction visualization (t-distributed stochastic neighbor embedding, t-SNE), considering temporal profiles of single-trial amplitude for unambiguous correct trials (i.e., trials with none of the distractors coinciding in space with the valid SC) suggested a larger difference across conditions for the low beta band (Fig 4B). We trained a random forest estimator to decode color conditions based on temporal profiles of either low or high beta amplitude using a subset of unambiguous correct trials. The decoder was tested on the remaining

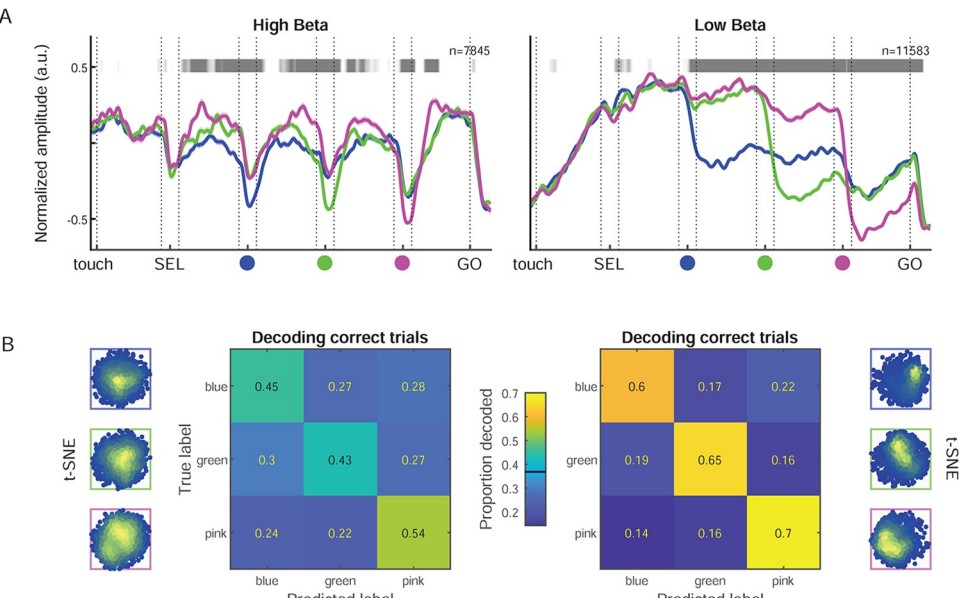

**Fig 4. Beta amplitude modulations and condition decoding in correct trials.** (A) Representation of the trial-averaged temporal profile of normalized high (left; 21–29 Hz) and low (right; 12–20 Hz) beta amplitude (+/− SEM), separated by the 3 color conditions, for all correct trials. The horizontal gray line above each plot graph represents the time-resolved modulation in beta amplitude by the color condition. The significativity is represented as a gray-scale gradient (brightest gray for $p = 0.01$ and darkest gray for $p <= 1e-08$; white means non-significant). (B) External left and right. Dimensionality reduction visualizations (t-SNE) for correct trials in the respective color conditions. Center. Decoding performance of SEL (color condition) category in correct trials, using either high (left) or low (right) beta band amplitude. Performance is presented as proportions of the total number of trials of each category in the test set (totaling 1 for each row). The diagonal represents the true positive accuracy, and the off-diagonal values correspond to the proportions of trials of each category incorrectly assigned to another category. The estimated chance level (0.37) is marked on the color scale bar. For t-SNE and decoding only unambiguous trials were included, in which none of the distractors coincided in space with the valid SC. Source data are available in S1 Data. SC, spatial cue; SEL, selection cue; t-SNE, t-distributed stochastic neighbor embedding.

unambiguous correct trials. The decoding performance was significantly above chance level for both bands, but strongest for the low band (Fig 4B).

This strong condition selectivity furthermore prompted us to explore the numerous distractor error trials (see S1 Table). Indeed, when the monkey wrongly performed a reach towards one of the distractors, the trial-averaged temporal profiles of low and high beta amplitude reflected the distractor selected by the animal (Fig 5A). For example, when the monkey selected the green distractor in blue trials (Fig 5A, top right, green curve), low beta amplitude had an average temporal profile similar to that for correct green trials (Fig 5A, middle right, green curve). The decoder trained on correct trials could also decode the attended distractor in error trials (Fig 5B) again with a better accuracy for the low band. In comparison, for both bands, the decoder was well below the chance level for decoding the missed valid SC in distractor error trials (Fig 5B). Thus, the temporal profile of both low and high beta band amplitude modulations in single trials reflected the behavioral choices made by the animal, whether correct or wrong.

## Trial-by-trial correlations with task-related and spontaneous behavior differ for low and high beta

We next investigated which task-related and spontaneous behaviors explained trial-by-trial amplitude variability of each beta band (Fig 6A) in a time-resolved manner. We adopted a

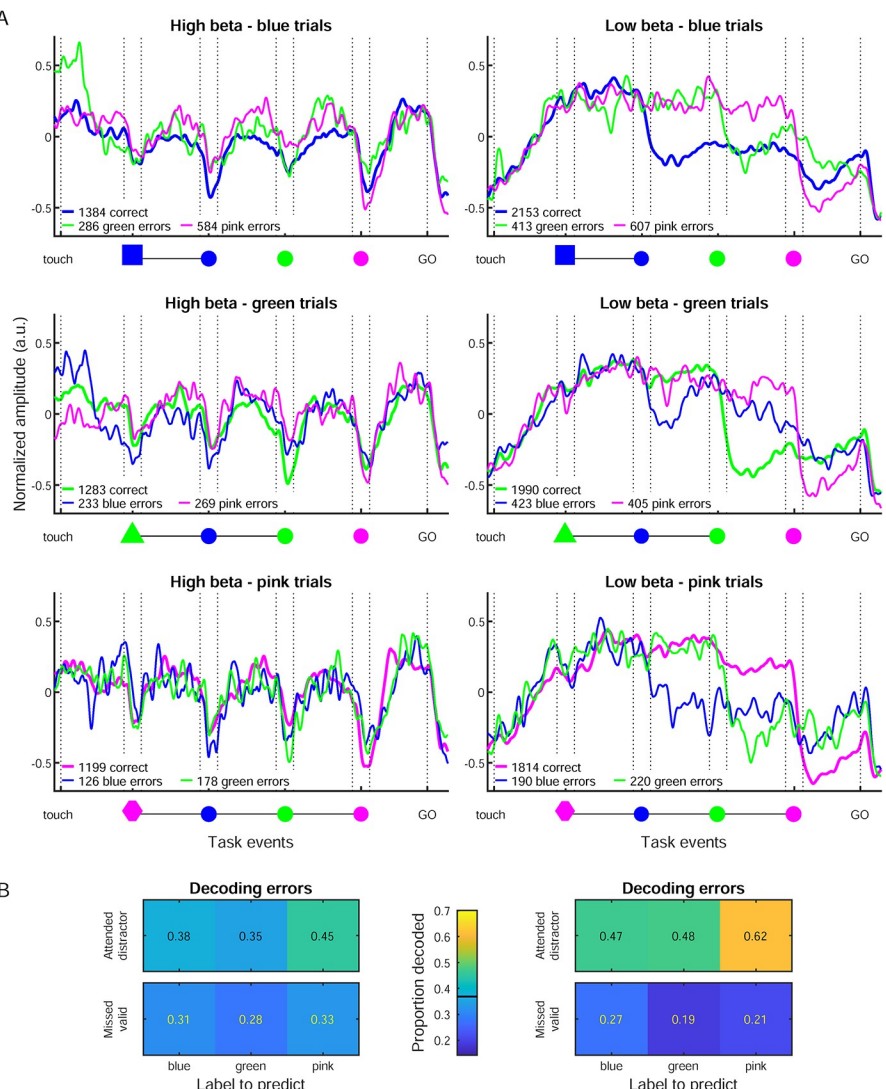

**Fig 5. Beta amplitude modulations and condition decoding in error trials.** (A) Trial-averaged beta amplitude in high beta (left) and low beta (right) in correct and distractor error trials, split for the 3 color conditions from top to bottom. Only unambiguous trials were included, in which the selected SC (whether correct or distractor) did not coincide in space with any of the 2 other SCs. Thicker lines represent correct trials and thinner lines the error trials in which either one or the other distractor was used. (B) Decoding performance in distractor error trials, using the classifier previously trained on the (unambiguous) correct trials. The first row for each beta band represents the accuracy when predicting the attended distractor (i.e., what the monkey actually did); the second row represents the accuracy when predicting the correct SEL category (i.e., what the monkey should have done). The same chance level applies to these predictions as for the decoding of correct trials in Fig 4 (0.37). Source data are available in S1 Data. SC, spatial cue; SEL, selection cue.

linear model (LM) approach in a time-resolved manner across the different trial epochs. The task-related factors encompassed the color condition, the RT and the direction of the upcoming reaching movement. The spontaneous (uninstructed) variables included hand micro-movement velocity, eye velocity, and gaze position. We here only consider gaze In/Out of the work area, since preliminary analyses revealed no systematic modulation of beta amplitude for gaze towards different items within the work area. We also included the time-on-task as a regressor. The comparisons of 247 models (all combinations of these 7 regressors and their

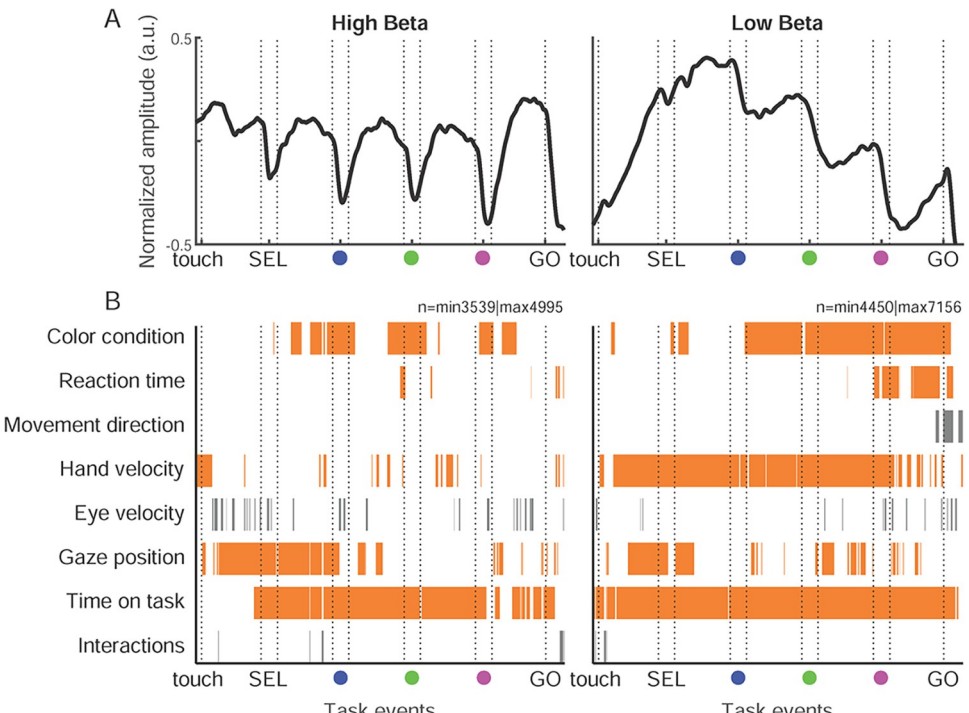

**Fig 6. Average low and high beta amplitude, and main regressors explaining amplitude variance.** (A) Representation of the trial-averaged temporal profile of normalized high (left; 21–29 Hz) and low (right; 12–20 Hz) beta amplitude (+/− SEM), for all correct trials with all color condition combined ($n$ = 7,845 for high beta, $n$ = 11,583 for low beta). The horizontal gray line above each plot graph represents the time-resolved modulation in beta amplitude by the color condition. The significativity is represented as a gray-scale gradient (brightest gray for $p$ = 0.01 and darkest gray for $p$< = 1e-08; white means nonsignificant). (B) Time-resolved representation of the presence of each regressor in the winning model after the application of a BIC for the comparison of all possible models and their 2-by-2 interactions, for the high beta (left) and the low beta (right). Each row represents a regressor, the last row represents all possible interactions. Regressors selected for ulterior analysis in orange and discarded regressors in gray. Source data are available in S1 Data. BIC, Bayesian index criterion; SEL, selection cue.

2-by-2 interactions) revealed that the winning model, with the lowest Bayesian information criterion (BIC) in each 10-ms bin along the trial, was mainly composed of a single or a combination of several non-interacting regressors (Fig 6B). Interaction terms were present in the winning models in only 19/1,360 bins (680 bins for each band). The direction of the upcoming movement was almost always absent in the winning model except around and after the GO signal for the low band. The eye velocity was also only sporadically part of the winning model. The results were remarkably similar for both animals (S4B and S4D Fig). Consequently, 5 LMs, each including 1 relevant regressor (color condition, RT, hand velocity, gaze direction, and time-on-task; excluding movement direction, eye velocity, and interactions), were fitted separately for the high and low beta bands. We chose to test each regressor separately, since in some sessions, one signal was not available or of poor-quality (mainly the eye signal). In a complementary analysis, we fitted for each color condition separately a multi-regressor model including the 4 other variables (RT, hand velocity, gaze position, and time-on-task). In this complementary analysis, the importance of each regressor in the full model was quantified by scrambling them across trials, one by one, and determining the loss in total variance explained in the full model with the relevant variable scrambled (S7 Fig). This complementary analysis largely confirmed the results described below.

The color condition explained trial-by-trial variability of both beta bands, also in this time-resolved analysis (Fig 6B). Importantly, the high band was modulated by the condition from SEL onset (sample) and strongest around each SC (valid versus distractor). The low band was modulated by the condition only from the SC1 (blue) onset, due to the selective drop in amplitude after each valid SC.

## Low beta reflects movement preparation and spontaneous postural dynamics

We then targeted the regressors identified as explaining mainly the low beta amplitude, namely RT and hand micro-movements (Fig 6B).

The RT did not affect the high beta band (Fig 7A), but correlated positively with low band amplitude, starting from each valid SC and up the GO signal (Fig 7B). Thus, from the onset of the cue that instructed the future movement, low beta amplitude reflected the level of preparation of the upcoming reaching movement.

Hand velocity strongly explained the trial-by-trial variability of low beta amplitude (Fig 7C–7E, vertical bar to the right). Low beta amplitude and hand micro-movement velocity correlated negatively during large portions of the trial. The correlation was reduced around the presentation of each valid SC, and in the final delay between SC3 (pink) and GO, in particular in the pink color condition (Fig 7E, right vertical bar). Despite the strong individual correlations of RT and hand velocity with low beta amplitude, the 2 behavioral factors were only sporadically correlated along the trial. Furthermore, a control analysis revealed that the variance in low beta amplitude explained by RT was not redundant with that of hand velocity (S8A Fig).

To better understand the nature of this strong relationship between low beta amplitude and hand micro-movement velocity during the maintenance of central hold, we performed a cross correlation analysis (Fig 7C–7E). This analysis confirmed the strong negative correlation between hand velocity and low beta amplitude. It furthermore showed the temporal dynamics of these correlations. In the early trial epochs up to valid SC onset, beta was *lagging* the hand by 120 to 130 ms. After valid SC onset, when low beta amplitude dropped (Fig 4A), and subsequently hand velocity increased (Fig 1C), cross correlations exhibited a wider pattern, with maximal negative correlation for 220 to 330 ms in the direction of beta *leading* hand. This lag is comparable to the latency difference of 260 to 370 ms between the moment of the steepest slopes of the average beta amplitude decrease and the subsequent average hand velocity increase. The correlation with beta lagging the hand by 120 to 130 ms remained, discernible as a "shoulder" in the cross correlograms (Fig 7F–7H, top).

## High beta reflects dynamical visuospatial attention

We next explored to which degree eye gaze direction (In/Out of the work area) affected beta amplitude (Fig 8). We quantified the correlation between gaze position (In/Out) and high and low beta amplitude at various temporal lags (S9A and S9B Fig). A lag with gaze *leading* beta by 240 ms resulted in the largest number of correlated bins across both bands. At this "optimal" lag, high beta amplitude was strongly affected by gaze position particularly in the trial epochs preceding the valid SC onset (i.e., different across the 3 conditions), and again just before the GO signal (Fig 8A), with much stronger amplitude for gaze In. During valid SC, high beta amplitude was similar for gaze In/Out. Thus, the characteristic modulation in average high beta amplitude by the "rhythmic" nature of the task was largely abolished when considering only gaze Out. Gaze position also influenced low beta band amplitude (Fig 8B); however, the effect was less strong. The most consistent effect across the 3 color conditions was a delay in

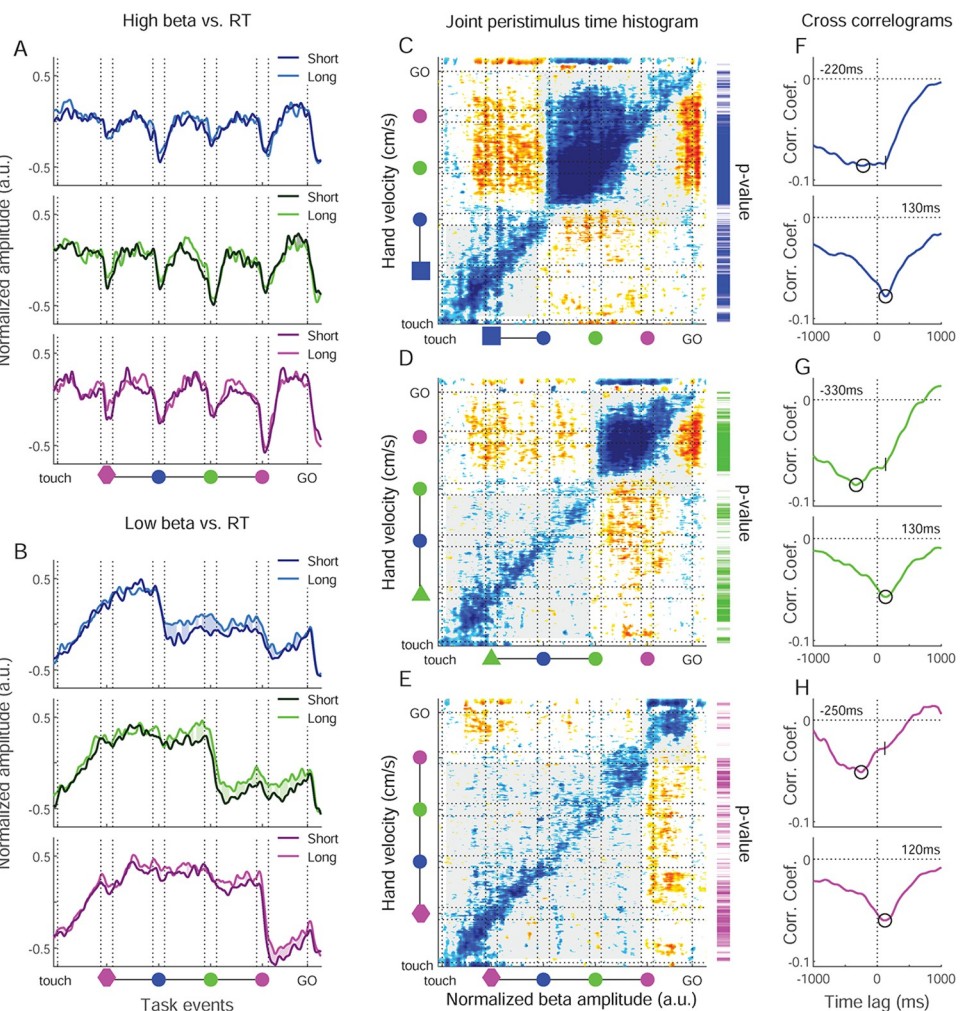

**Fig 7. Correlations of beta amplitude with RT and hand velocity.** (A) High beta amplitude split into 2 groups based on RT, for the quartiles of trials with longest (bright curves) and shortest (dark curves) RT in each session, for each color condition separately. Colored areas between the 2 curves indicate correlation significance (and sign) between high beta amplitude and RT, calculated across all trials. The hues of the shaded areas represent the sign of the significativity. Brighter hues represent positive correlations and darker hues negative correlations between beta amplitude and RT. (B) Same representation for the low beta band. (C–E) Left: Equivalent of a joint peristimulus time histogram (jpsth) applied to the hand velocity and the low beta amplitude along the trial. Each point of the matrix represents the corrected trial-by-trial cross product of the 2 variables on 10-ms bins. The analysis was performed separately for the 3 color conditions. Each colored matrix point was inferior (cold color) or superior (warm color) to all 100 values from shuffled matrices (equivalent *p*-value of 0.01). The vertical and horizontal lines represent the appearance and disappearance of the valid SC for the 3 conditions. Gray shaded regions represent the trial periods considered in the corresponding cross-correlogram. Vertical color bars on the right of each jpsth represent the significativity of the correlation between low beta and hand velocity, for each color condition. The significativity is represented as a color-scale gradient (brightest color for *p* = 0.01 and darkest color for *p*< = 1e-08; white means nonsignificant). (F–H) Cross correlograms for the 3 color conditions. Each value of the cross correlogram represents the average of main and para diagonals of the matrices, separately for before (bottom) and after (top) the onset of the valid SC. The circles on each curve represent the most negative value of the correlation. On the top plots (post-validSC), the vertical dash represents the moment of the most negative value in the pre-validSC period. Source data are available in S1 Data. RT, reaction time; SC, spatial cue.

the characteristic drop of low beta amplitude after the valid SC onset, for gaze Out. For both bands, the effect was much reduced or absent when considering other lags between gaze and beta amplitude (S9D and S9E Fig). For the low band, we furthermore split trials according to

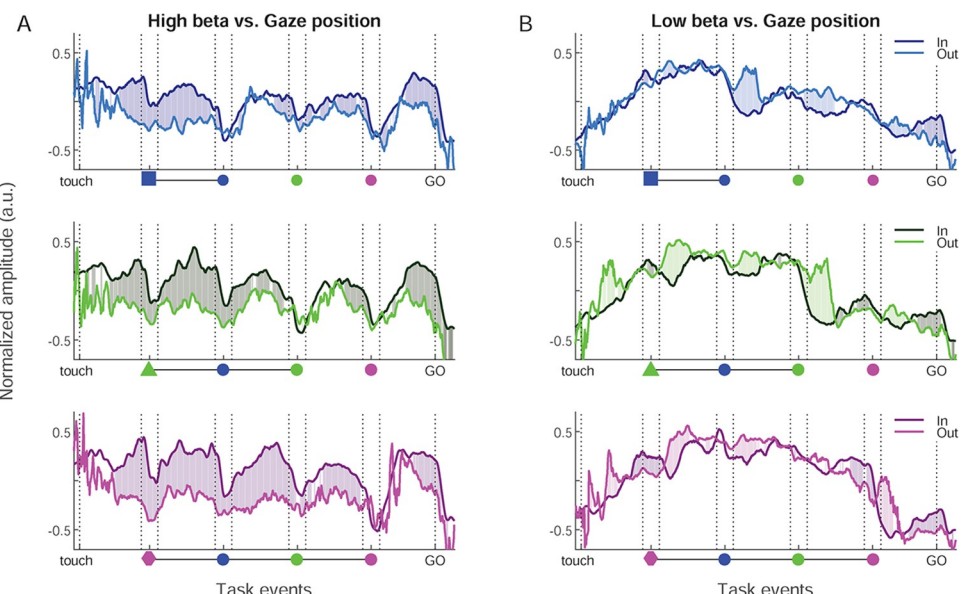

**Fig 8. Correlation of high and low beta amplitude with gaze position.** (A) High beta amplitude split in 2 groups based on gaze position, for each color condition. Dark curves for gaze In, and bright curves for gaze Out, in each time point for the beta amplitude considering the gaze position exactly 240 ms before. The significativity is represented in the same way as in Fig 7A and 7B. (B) Same representation for the low beta band. Source data are available in S1 Data.

gaze position taken at exactly 200 ms after the valid SC onset. This confirmed that the drop in beta amplitude after the valid SC mainly occurred for trials with gaze In, compared to trials with the gaze still Out (Fig 8C).

## Time on task

Finally, for both beta bands the amplitude increased systematically with time-on-task (Fig 9). A spectral parametrization analysis confirmed that amplitude increases were specific to the 2 beta bands and not caused by a change in the overall level or the slope of the aperiodic signal, which remained unchanged (S10 Fig). For the high band, the amplitude increase for late trials occurred around SEL, and strongly and selectively in the delay immediately preceding each valid SC. For the low band, the amplitude increased within the session across most trial epochs, both before and after valid SC onset. We described above how RT was positively correlated with time-on-task, and furthermore that RT correlated positively with low beta amplitude between valid SC and GO. A control analysis revealed that the variance in low beta amplitude explained by RT described above was in part redundant with that of time-on-task (S8C Fig).

## Discussion

Sensorimotor beta rhythms remain enigmatic, despite 75 years of scientific efforts since Jasper and Penfield [1] first described their association with voluntary movement. We here describe a double-dissociation in sources and behavioral correlates of motor cortical low and high beta, with respect to both task-instructed and spontaneous behavior. In 2 macaques performing a delayed visuomotor reaching task, low beta dominated in M1, while high beta dominated in PMd. Low beta correlated positively with RT during preparation and negatively with unin-structed hand postural micro-movements throughout the trial. In contrast, high beta was unrelated to RT and hand postural dynamics, and instead modulated selectively in anticipation of and during visual cues, reflecting rhythmic temporal predictions. However, this rhythmic

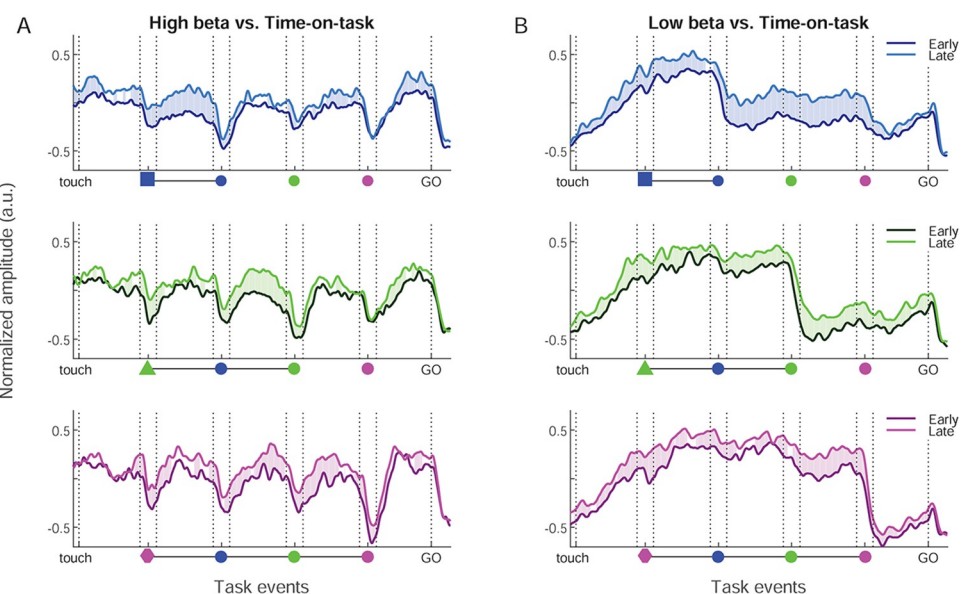

**Fig 9. Increase in high and low beta amplitude with time-on-task.** (A) High beta amplitude split in 2 groups based on time-on-task, for each color condition. Dark curves for the first third of trials and bright curves for the final third of trials in each session. The significativity is represented in the same way as in Fig 7A and 7B. (B) Same representation for the low beta band. Source data are available in S1 Data.

modulation was largely abolished when focal attention was oriented away from the work area. Our clear-cut findings reconcile the largely disparate roles proposed for the broader beta band (approximately 13 to 35 Hz) in the motor cortex [23], designating specific roles in movement control for M1 low beta and spatiotemporal attention for PMd high beta.

## Spontaneous hand and eye movements reflect movement preparation and focal attention

The monkeys performed a delayed match to sample task entailing strong working memory components, first for selecting the valid spatial cue (target) based on color matching, and second to memorize the target position while preparing the reach. Both monkeys performed spontaneous (uninstructed) hand and eye movements that were aligned to the task events and specific for each color condition (S2 Fig). Such spontaneous movements were shown to persist even in highly constrained settings [33]. Our monkeys had to maintain their hand position within a very limited zone through most of the trial. Yet, they frequently made micro-movements with the hand, in particular during movement preparation. These spontaneous movements did not reflect the planned movement direction, but were possibly related to increased postural muscle tonus during movement preparation being imperfectly balanced across different muscles.

Furthermore, our monkeys were head-fixed, but free to move their eyes. They frequently shifted their gaze between items in the visual scene, or In and Out of the work area. We interpret these uninstructed shifts of gaze In/Out as reflections of spontaneous switches between covert attention (gaze Out) versus overt or focal attention (gaze In). In our task, trial start and GO were more than 6 s apart, which is very long for maintaining focal overt attention. Having the gaze out of the work area, which was more frequent before the valid cue presentation, probably reflected having less focal attention on the visual scene. The "rhythmic" temporal predictability of the task events permitted shifting the gaze to the work area in anticipation of

or triggered by the salient visual events, in particular the valid cue. This probably explains why the performance was correct even with periods of the trial being monitored with peripheral vision and covert attention.

## Low beta dominates in M1 and high beta dominates in PMd

We found low beta to dominate in M1, while high beta dominated in PMd. Rather than a gradual shift in peak frequency of a single beta band along the posterior-anterior axis, we observed 2 distinct bands also at intermediate sites. Most studies of sensorimotor beta rhythms in monkeys lumped frequencies from approximately 13 to 35 Hz, such that any gradient across cortex might have been overlooked. However, consistent with our result, Chandrasekaran and colleagues [30] found beta peak frequency in PMd to be above 20 Hz. Furthermore, near the central sulcus (including M1 and somatosensory areas) beta was mainly observed to peak at or below 20 Hz [47–52] (but see also [4,11]). Several studies of beta rhythms in monkey prefrontal cortex reported peak frequencies above 20 Hz [39,40,53–55]. Consistent with this, Vezoli and colleagues [41] found high beta to be dominant anterior to low beta across the fronto-parietal cortex in macaque ECoG. In humans, Rosanova and colleagues [56] showed that single-pulse transcranial stimulation induced bursts of low beta in parietal cortex and high beta in premotor cortex, and Mahjoory and collagues [57] reported a gradual increase in beta peak frequency along the posterior-anterior axis in resting state magnetoencephalography. It is therefore probable that the low and high beta bands that we here characterize across motor cortex extend well beyond, and at the cortical level reflect a low beta network including M1, somatosensory and parietal regions and a high beta network including PMd and prefrontal regions, not excluding further involvement of also subcortical structures such as the basal ganglia [46,48,58].

The distinctly different temporal profiles of M1 low and PMd high beta amplitudes were mirrored in the very different temporal profiles of average MUA amplitude when split according to beta band dominance. This suggests a functional split across the motor cortex, with band-specific roles in movement control and spatiotemporal attention, respectively. Note however that individual neurons have very diverse activity profiles in such visuomotor tasks [27], and some rather inverse to the average MUA amplitude profile. Furthermore, there is no trial-by-trial correlation between the firing rate of individual neurons and beta amplitude [28].

## Low beta reflects movement preparation and continuous postural dynamics

Many studies described a decrease in sensorimotor beta amplitude during movement preparation [23], but here we show this to be restricted to the low band, following the valid spatial cue. Lack of focal attention at the time of the valid cue onset (gaze Out) delayed this amplitude drop, and possibly also the onset of movement preparation. Along with the negative correlation with spontaneous hand micro-movements and the positive correlation with RT, these are strong evidences for M1 low beta reflecting movement preparation and postural control processes. In contrast, the amplitude of the high beta band in PMd was largely independent of hand micro-movements and behavioral RT, and remained high during movement preparation. The drop in average low beta amplitude after valid cue onset was accompanied by an inverse modulation in the average amplitude of local MUA, which increased after valid cue onset. This underscores the importance of low-beta dominant cortical sites in hand movement control. As low beta was so strongly correlated with the spontaneous hand micro-movements that changed across color conditions (S2 Fig), the ability to decode color conditions using the temporal profile of low beta might partly be attributed to the hand micro-movements.

Time-on-task effects were attributed to increased cognitive effort or decreased vigilance in attentionally demanding tasks [29,42]. Consistent with this, we found that behavioral RT increased with time-on-task. The RT was furthermore positively correlated with low beta amplitude, but only during movement preparation, not prior to the valid spatial cue presentation. Low beta amplitude was also positively correlated with time-on-task, across all trial epochs. A control analysis revealed that RT was partly redundant with time-on-task in explaining low beta amplitude. As RT only affected low beta amplitude during movement preparation, taken together these results suggest the time-on-task effects reflected increased cognitive effort related to postural control and movement preparation, rather than a nonspecific decrease in vigilance.

A recent study reported transient beta bursts even during sustained isometric gripping in humans [59], suggesting no direct link between cortical beta amplitude and motor output. The spontaneous hand postural micro-movements we observed were 100-fold smaller in velocity than the center-out reaching responses (S1A Fig), yet strongly correlated with low beta amplitude across all trial epochs. The discrepancy between their and our findings might be due to the large number of trials available for our analysis, or less sensitivity in their setup for detecting minute changes in grip force during the sustained isometric contraction.

The negative correlation between low beta and hand velocity was maximal for beta lagging hand by about 120 to 130 ms prior to the valid cue (Fig 7F–7H). In comparison, directed descending and ascending coherence in the beta frequency range between cortex and muscle [60,61] were reported to have much shorter phase delays than this, on the order of 25 ms. Instead, Jasper and Penfield [1] already speculated whether the emergence of sensorimotor beta bursts reflects a "state of equilibrium of activity permitting again a synchronization of unit discharge." This could be an equilibrium between excitation and inhibition [62], with beta reflecting the network resonance frequency [56,57,63–65]. In this view, a decrease in M1 low beta amplitude *lagging* an increase in hand micro-movement velocity might reflect a shift away from neuronal population activity equilibrium, with the explicit aim to prevent the hand cursor from sliding outside the central fixation spot, since this would abort the trial.

During movement preparation, the correlation with beta lagging hand persisted, but low beta amplitude and hand velocity here correlated negatively across a broad range of lags, with maximal correlation strength for beta *leading* the hand of 220 to 330 ms. This lag is close to, and presumably dominated by, the average temporal difference of 260 to 370 ms between the drop of low beta and the subsequent increase in hand velocity.

## High beta reflects temporal task prediction and focal attention

Beta amplitude modulations scale to predictable delay durations in visuomotor tasks [20,22]. We found that whereas the condition selectivity in the low band only emerged during movement preparation, the high band was already selective before the first spatial cue, in a predictive manner. Beta amplitude in frontal cortex was found to modulate with "rhythmic" visuomotor or working memory tasks [18,39] and passive auditory tasks [19] permitting rhythmic temporal predictions. We here demonstrate that rhythmic beta amplitude modulations related to temporal prediction and attention were restricted to the high band and were much enhanced during focal attention towards the work area (gaze In), in particular, in the delays leading up to the presentation of the valid cue. Since high beta was correlated with gaze position that changed across color conditions (S2 Fig), the ability to decode color conditions using the temporal profile of high beta might partly be assigned to the gaze dynamics.

The rhythmic amplitude modulations we here describe for PMd high beta strongly resemble those reported by Lundqvist and colleagues [39] for macaque prefrontal cortex high beta

during a working memory task requiring central eye fixation (i.e., comparable to *gaze In* for our study) and saccade responses. A phase locking analysis confirmed that the high beta band we observed in the PMd sites was locally generated. Furthermore, the average amplitude of local MUA in high beta dominant sites modulated inversely to the high beta amplitude (as were also gamma bursts and neuronal firing rates in [39]), and peaked during the processing of valid visual spatial cues. The similarity of high beta amplitude modulations for these 2 experimental settings suggests a general role in effector-independent spatiotemporal prediction and attention in frontal cortex. High beta amplitude increased with time-on-task, but mainly during SEL cue presentation and in a condition-dependent manner in the delay immediately preceding and during the valid cue. This might reflect increased allocation of spatiotemporal attention towards the most relevant visual cues (SEL and valid SC), as the animal fatigued [29,42].

## Materials and methods

### Experimental design

**Animal preparation.** Two adult male Rhesus monkeys (T and M, 10 and 14 kg, respectively) participated in this study. Care and treatment of the animals during all stages of the experiments conformed to the European Commission Regulations (Directive 2010/63/EU on the protection of animals used for scientific purposes) applied to French laws (decision of the 1st of February 2013). The experimental protocol was evaluated by the local Ethics Committee (CEEA 071) and carried out in a licensed institution (B1301404) under the authorization 03383.02 delivered by the French Ministry of High Education and Research. Previously published studies using data from these 2 monkeys [20,27,28,66–68] were based on recordings from the opposite hemisphere during performance of another visuomotor task. The 2 macaques used in this study were monitored daily, either by the animal care staff or the researchers involved in the study. The facility veterinary controlled regularly the general health and welfare conditions of the animals. The animals were pair-housed, and toys and enrichment, usually filled with treats, were routinely introduced in their home cage to promote exploratory behavior. During task performance, the animals received liquid reward from a dispenser. The animals were water-restricted in their home cage, with free access to dry pellets. In the event of reduced liquid consumption during task performance, the minimum daily intake was reached by giving extra water and fruit or vegetables in the home cage, delayed for a few hours after the end of training. The daily fluid intake was never below 18 ml/kg, a low level for which it has been shown that macaques are able to effectively modulate their blood osmolality [69], based on each animal's reference body weight (measured prior to entering the liquid restriction regime). On resting days (e.g., weekends), the animals received a complete ration of liquid in the form of water and fruits in the home cage.

Subsequent to learning the visuomotor task (see below), the monkeys were prepared for multi-electrode recordings in the left hemisphere of the motor cortex (M1 and PMd), contralateral to the trained arm. In a first surgery, prior to completed task learning, a titanium headpost was implanted posteriorly on the skull, fixated with titanium bone screws and bone cement. In a second surgery, several months later, a cylindrical titanium recording chamber (19 mm inner diameter) was implanted. The positioning of the chamber above upper-limb regions of M1 and PMd was confirmed with T1-weighted MRI scans (prior to surgery in both animals and also postmortem in monkey M), and with intra-cortical electrical microstimulation (ICMS; as described in [70]) performed at the end of single-tip electrode recording days in the first recording weeks, in both monkeys. The recording sites included in this study spanned about 15 mm across the cortical surface in the anterior-posterior axis and only

include sites determined with ICMS to be related to upper limb movements (Fig 3C). The exact border between PMd or M1 areas was not estimated.

**Behavioral setup and task.** The 2 monkeys were trained to perform a visuomotor delayed match to sample task with fixed cue order and a GO signal (Fig 1B). The task required arm-reaching responses in one of 4 (diagonal) directions from a common center position, performed by holding a handle that was freely movable in the 2D horizontal plane. The visual scene was displayed on a vertical computer monitor (LCD; 75 Hz) in front of the monkey (Fig 1A). We here describe the monitor stimuli in cm units, but since the viewing distance was about 57 cm, this approximates to the same degrees of visual angle. Before the start of each trial, the monitor displayed the handle (hand cursor) position (small white square; 0.4 cm edges), a central fixation spot (yellow flashing disc; 0.45 cm radius), and the 4 possible peripheral target positions (red circular outlines; 1.5 cm radius at 9 cm diagonal distances from the center). The position of the cursor was updated on the monitor every 40 ms (approximately every third frame), but only if the accumulated displacement from the previous update exceeded 0.1 cm (to avoid flicker due to electronic noise).

The monkey initiated the trial by positioning the cursor inside the central hand fixation spot. This central touch ended the flashing of the fixation spot (which remained on) and was accompanied by an auditory tone, presented for 50 ms. After holding this central position for 1,000 ms, a selection cue (SEL) indicating the color to attend for that trial appeared on the screen for 300 ms, displayed behind but extending well beyond the central yellow disc and the overlying hand cursor. SEL consisted of one out of 3 differently colored polygons (blue, green, or pink; approximately 3-cm radius) defining the color condition. A 1,000 ms delay followed SEL offset. Thereafter, 3 peripheral spatial cues (SC1-3) were presented in sequence, each displayed for 300 ms, with 1,000 ms delay after each of them. The SCs were colored discs (0.9 cm radius), always presented in the temporal order blue-green-pink, each within one of the 4 peripheral red outlines.

All 4 diagonal target positions were equally likely for each SC. Thus, successive SC in the same trial could be presented in the same position. This resulted in 192 unique trials, combining the 3 color conditions with the 4 independent positions for SC1, SC2, and SC3. In monkey T, who was not willing to work for as many trials as monkey M, only 3 of the 4 target positions were used in each session (randomly selected for each session), to reduce the number of unique trials. For both animals, to ease the task, the 3 color conditions were presented separately in small blocks of approximately 15 unique trials per block, cycling across multiple blocks of the 3 color conditions to complete all the unique trials. The unique trials within each block were presented in pseudo-random order. Incorrect trials within a block were re-presented later in the same block, and each block was completed only when all unique trials in the block were correctly executed.

The animal had to select the (valid) SC according to the color indicated by SEL (i.e., delayed color match to sample) and ignore the 2 (distractor) SCs of different colors. The GO signal was presented after the final 1,000 ms delay following SC3, prompting the animal to execute the center-out arm reaching movement to the memorized valid SC position. The GO signal was directionally non-informative, consisting in the simultaneous onset of 4 red light-emitting diodes (LEDs; embedded in a thin Plexiglas plate in front of the monitor) at the centers of the 4 circular target outlines. The RT and movement time each had a maximum allowance of 500 ms. The animal was trained to stop and "hold" within the correct peripheral target outline for 300 ms to obtain a reward. The touch of the valid target was signaled by an auditory tone (50 ms) and a completed hold with another tone (50 ms). Reward was delivered 500 ms after completed hold and consisted in a small drop of liquid (water or diluted fruit juice). Monkey T was not rewarded for non-hold trials, while monkey M was given a smaller reward on non-hold

trials (on the valid target; 500 ms after breaking hold). For both animals, these non-hold trials were included in the analysis (about 10% of all included trials).

The manual (horizontal) work area of the monkey was scaled down with respect to the display on the monitor (by a factor of about 0.7). In manual (horizontal) space, the diagonal distance (center to center) between the fixation spot and peripheral targets was 6.5 cm. The required central fixation zone was defined to be within a radius of 0.3 cm, and the accepted touch zone of the peripheral targets had a radius of 1 cm. These touch zones corresponded to the hand cursor overlapping more than halfway with the fixation spot or the peripheral outlines. In the offline analysis of the hand signal, we used the spatial scaling of the visual scene on the computer monitor.

In short, in this delayed match to sample task, the timing and sequential order of the 3 SCs were fully predictable, and SC validity was cued at the start of each trial by SEL. Only the spatial positions of the 3 SCs were unpredictable.

**Data acquisition.**   During recording days (maximally 5 days a week), a multi-electrode, computer-controlled microdrive (MT-EPS, Alpha Omega, Nazareth Illith, Israel) was attached to the recording chamber and used to transdurally insert up to 5 single-tip microelectrodes (typical impedance 0.3 to 1.2 MΩ at 1,000 Hz; FHC) or up to 2 linear microelectrode arrays (either V- or S-probes, Plexon, Dallas, Texas, United States of America or LMA, Alpha Omega; each with 24 or 32 contacts, inter-contact spacing either 100, 150, or 200 μm; 12.5 or 15 μm micrometer contact diameters) into motor cortex. In this study, we employ the term "site" for the recording obtained from each individual single-tip electrode (or from each linear array) recorded in individual behavioral sessions. The electrodes (or arrays) were positioned and lowered independently within the chamber (Flex-MT drive; Alpha Omega) in each session. Individual guide-tubes for each electrode/array were used that did not penetrate the dura (no guide was used for the more rigid LMA array). For single-tip electrodes, the reference was common to all electrodes and connected, together with the ground, on a metal screw on the saline-filled titanium recording chamber. For the linear array recordings, the reference was specific to each array type. S2 Table summarizes the different reference positions used. For the LMA (Alpha Omega), it was an insulated wire exposed at the tip, either emerged in the chamber saline, or attached with a crocodile clip to the probe stainless steel tube (which in turn was lowered into the chamber liquid, but not extending into brain tissue, as the lower part of the probe was epoxy-insulated). For the V- and S-probes (Plexon), in most cases the reference was the stainless steel shaft of the array (extending into brain tissue, in near proximity to the probe's recording contacts). In a few sessions, the reference was instead placed on a skull-screw on the more posterior headpost (6/36 sites using V-probes in monkey T) or on a screw on the saline-filled recording chamber (2/50 sites using S-probes in monkey M). For both array types, the ground was either connected to a skull-screw of the remote titanium head-fixation post or to a screw of the titanium recording chamber.

We used 2 different data acquisition (DAQ) systems to record neuronal and behavioral data. All single-tip electrode recordings in monkey T were obtained on a recording platform with components commercialized by Alpha Omega. This system included the Alpha-Map system for online monitoring of signals (running on Windows XP), and the MCP-Plus multi-channel signal processor including analog head-stages. Neuronal signals from each electrode were amplified with a gain of 5,000 to 10,000 (with unit-gain head-stage), hardware filtered (1 Hz to 10 kHz), and digitized and saved for offline analysis at a sampling rate of 32 kHz.

All linear array recordings in monkey T, and all recordings (single electrodes and linear arrays) in monkey M, were obtained on a recording platform with components commercialized by Blackrock Neurotech (Salt Lake City, Utah, USA). This system included Cereplex M digital head-stages (versions PN 6956, PN 9360, and PN 10129) connected to a Digital Hub

(versions PN 6973, PN 6973 DEV 16–021, and PN 10480) via custom HDMI cables (versions PN 8083 and PN 8068), which transmitted signals via fiber optics to a 128 channel neural signal processor (NSP hardware version 1.0), and control software Cerebus Central Suite (v6.03 and v6.05 for monkeys T and M, respectively; running on Windows 7). An adapter (PN 9038) permitted connecting multiple single-tip electrodes to the Cereplex M Omnetics connector (Monkey M). Neuronal signals were hardware filtered (0.3 Hz to 7.5 kHz) and digitized and saved for offline analysis at a sampling rate of 30 kHz.

Behavioral event codes (TTL, 8 bits) were transmitted online to the DAQ system from the VCortex software (version 2.2 running on Win XP; NIMH, http://dally.nimh.nih.gov), which was used to control the behavioral task. A custom rebuild of the VCortex software allowed simultaneous online monitoring of hand and eye gaze positions in the common reference frame of the animal's visual monitor display. Continuous hand position (X and Y) was obtained from 2 perpendicularly superimposed contactless linear position magnetorestrictive transducers (model MK4 A; GEFRAN, Provaglio d'Iseo, Italy). The "floating" magnetic cursor was attached to a manipulandum that could be moved along 2 pairs of rails with ball bearings, each pair aligned with one of the 2 transducers. The Y-oriented rails were fixed on top of the X-oriented rails. As such, this system provided somewhat less frictional resistance in the Y direction than in the X direction. Furthermore, either of the uni-directional X or Y displacements provided somewhat less frictional resistance than their combination needed to move to the diagonally placed targets. Hand position was used online to control the behavioral task. The hand position was also saved by VCortex for offline analysis (at 250 Hz sampling rate). In a majority of sessions, eye gaze position (X and Y) was recorded by the DAQ system (video based infrared eye-tracking; RK-716PCI (PAL version) at 50 Hz for the first single-tip electrode recordings in monkey T, or ETL-200 at 240 Hz sampling rate for the array recordings in monkey T and all recordings in monkey M; ISCAN Inc., Woburn, Massachusetts, USA). The eye-tracking camera was positioned next to the lower right corner of the monkey's computer monitor.

In many sessions, we also recorded heart rate (plethysmographic pulse waveform from ear-clip pulse oximeter, model 8600V; Nonin Medical Inc., Plymouth, Minnesota, USA), and in some sessions, surface electromyogram (EMG) from 1 or 2 proximal upper limb muscles (deltoid/biceps).

### Statistical analysis

**Behavioral performance.**   All analyses of behavioral and neuronal data were conducted offline by using Matlab (The MathWorks, Inc.) and Python. Multiple comparison chi-squared tests using the Matlab function crosstab were used to compare percentages of distractor errors for the 3 color conditions and the 4 movement directions; 3-way ANOVAs were used to quantify the variability in RT across sessions, color conditions, and movement directions for each monkey. Finally, to determine whether there was any systematic within-session modulation of RT, we normalized (z-scored) the RT within each session (to compensate for any differences in average RT across sessions), before collapsing trials across all sessions for each monkey. We then calculated the Pearson correlation coefficient between RT and trial number inside each session.

**Hand position analysis.**   The hand position signals that were recorded with VCortex were realigned in time with the other data recorded by the DAQ system offline, by realigning the behavioral event codes and up-sampled (linear interpolation) from 250 Hz to 1 kHz. The hand position signals were calibrated (scaled) online in the VCortex configuration to match the

visual display before storing on file, and in analysis, we used the spatial scaling of the visual scene in cm.

The RTs for the center-out reaching movements were redefined offline using the hand trajectories. First, hand velocity and acceleration were computed in each trial, using a Savitsky–Golay algorithm. To determine reach movement onset, in a 2,000 ms duration epoch centered on GO, periods with prolonged increased velocity (>50 ms) above an empirically determined velocity-threshold (6 cm/s) were then detected, and the final, preceding increase in acceleration above an empirically determined acceleration-threshold (6 cm/s/s) was then taken as the time of movement onset. These RTs were confirmed in both animals by visual inspection of single trial trajectories in several sessions.

We also quantified hand micro-movements during the maintenance of stable central hand position using hand velocity and position.

**Eye position offline calibration and analysis.** In a majority of sessions, we recorded eye position with an infrared camera. A rough online calibration of the gain and offset of the eye X and Y signals were done during the first behavioral trials in each recording session, to compensate for small changes in head fixation or camera position compared to the previous day/session. This simplified online calibration was adopted to avoid training the monkey in a fixation task. The center of gaze was set to zero (center) while the monkey looked at the small yellow central target in order to place the hand cursor therein to initiate a new trial. Then, on some days the X or Y gain was updated slightly so that the spontaneous eye fixations on the peripheral target outlines matched their position in the Cortex software interface. The trials before calibration (typically 0 to 3 correct trials) were excluded in offline analysis involving eye movements.

For data analysis, the eye signals recorded with the DAQ system were re-calibrated offline, to correct for the distortion induced by having the camera off the horizontal and vertical central axes of gaze. First, the raw eye signals were inspected visually to exclude from offline calibration and analysis the trials that were recorded before the completion of the rough online calibration, typically consisting in suppressing the 0 to 3 first correct trials in each session. Raw data were downsampled from the acquisition sampling frequency (1 or 30 kHz) to the camera sampling frequency (50 or 240 Hz) and linearly rescaled from bits to volts. We computed the eye velocity in volts/s using the Savitzky–Golay algorithm. For the offline calibration algorithm, we only considered data points that likely belong to fixation periods (i.e., whose velocity was lower than the lower 10th percentile of the total velocity distribution). At this stage, the superimposition of eye positions during these low velocity epochs across all trials in a session already showed an expected clustering of the data around 5 positions on the screen whose geometry resembled the center and 4 peripheral target positions used in the task. Thus, we were able to define boundaries in the voltage space to separate data points according to whether they were recorded when the monkey was looking within the work area (approximate boundaries of computer monitor) or when he was looking away from the work area (e.g., looking in the ceiling or signal saturation due to eye blinks). The low velocity (fixation) data occurring within the work area was then sorted into 5 clusters using a k-means algorithm (kmeans function in MatLab, using squared Euclidean distance). Cluster centers were assumed to represent the target positions in the voltage space. We next generated a 2D nonlinear model to compensate for the distortion due to camera position, between target coordinates on the screen (in cm) and voltage amplitudes of the corresponding centroids. This was achieved by adjusting a polynomial function to fit the relationship between each coordinate in the screen space to the XY coordinates in the voltage space. The correction was then applied to the complete eye traces. A detailed version of this correction can be found in [71]. Each data point was re-assigned to a cluster if it was located at a distance <2 cm from the target's center coordinates,

or assigned as being between clusters (but within the work area), or outside of the work area (incl. saturated). Eye position, velocity, and acceleration were then saved for further analysis, scaled in cm of the visual display, alongside cluster membership of each data point. Furthermore, the data points outside the work area that were beyond the lower or upper 0.99 percentiles of the boundaries of the raw X and Y voltage signals were marked as "saturated."

To detect the saccadic eye movements, we applied a recursive algorithm that seeks for the largest breakpoint in a piecewise stationary process, in a trial-by-trial fashion. First, we computed the cumulative 2D velocity of the eye signal in cm/s. This representation yields a pseudo staircase profile alternating between steep and slowly increasing periods over time. We extracted the highest decile of the velocity distribution and marked the corresponding steps in the staircase as boundaries to define periods when the subject was looking coarsely in the same area. These steps corresponded to blinks or to obvious large saccades and the steady periods were either fixation periods or multiple fixation periods with intermittent smaller saccades. During the steady periods, the cumulative distribution showed a slow increase due to noise originating from micro-movements and the recording device. The contribution of this noise being dependent on the location of the fixation on the screen, we compensated for it by subtracting the average slope for each period separately. This gave a piecewise stationary process that showed pseudo-horizontal steady epochs with better signal to noise ratio for the intermittent smaller saccades. Secondly, we applied a recursive algorithm to this process consisting, within a given time window, to compute at each data point the difference between the prior and the posterior average values. The maximum difference was extracted and compared to a threshold value computed after the velocity profile of a reference saccade (10 ms duration, 60 cm/s velocity peak). If the maximum difference was larger than the threshold, it was considered an actual transition and the time window was split in 2 at this time point. Starting with a time window covering the whole trial, the algorithm defined new (smaller and smaller) time windows at each iteration and the new window boundaries were considered as transitions. To avoid transitions to be detected multiple times, we introduced a "refractory period" of +/− 15 ms around accepted transitions. Fixation periods were finally defined by sorting the transitions between fixations into detected saccades or detected micro-saccades depending whether or not the Euclidian distance between the isobarycenter of 2 successive fixations was larger than a threshold (the change in eye position on the screen for an eye movement of 0.5 cm). Saccade onset/offset times were saved for further offline analyses alongside the other calibrated eye signals detailed above.

Finally, eyeblinks were detected as 2 subsequent (<150 ms apart) eye signal velocity passings beyond a velocity threshold (500 cm/s for the 50 Hz sessions and 800 cm/s for the 240 Hz sessions). The data points in a window including the gap between these subsequent threshold passings, as well as a couple of preceding and subsequent flanker data points were marked as eyeblinks. Visual inspection confirmed that this method was able to distinguish between saccades and abrupt velocity increases due to eyeblinks, even if large standalone saccades sometimes had velocities beyond the thresholds used for eyeblink detection.

**Decoding task condition with behavior.** We built 2 classifiers based on the temporal evolution of the hand velocity and the gaze position in single trials. In both cases, the signal was cut between SEL and GO and then downsampled by averaging in 50 ms non-overlapping windows in each trial. All correct trials in all sessions in which the specific behavioral signal was exploitable were included ($n$ = 11,587 for hand velocity and $n$ = 8,630 for eye gaze position). A random forest classifier was trained in 60% of the data and tested in the remainder 40%. This procedure was repeated in 20 different train/test splits to ensure stability of the results. The average confusion matrix across all runs was computed. The estimated chance level based on shuffling the data (100 shuffles per decoder) never exceeded 0.35.

**LFP spectral analysis and beta amplitude extraction.**    All sessions with sufficient quality of data were included in analysis. The raw signals were low-pass filtered offline at 250 Hz cutoff frequency (zero-phase fourth order Butterworth filter, using the butter and filtfilt functions in Matlab) to obtain the LFP signal, which was then downsampled to 1 KHz and saved for further analysis. For this study, we included only 1 contact for each of the linear array penetrations, selected to be well within cortex and with low noise (e.g., no heartbeat artifacts). LFP activity from 110 individual sites (63 with single-tip electrodes and 47 with linear arrays) in 59 sessions monkey T and 60 sites in 39 sessions (10 with single-tip electrodes and 50 with linear arrays) in monkey M were included in the analysis. A site is here defined as the conjunction of a specific chamber coordinate of the electrode entry and cortical depth, in one recording session. In the included LFP sites, trials with obvious artifacts (mainly due to teeth grinding, static electricity, or heart-beat signal) detected by visual inspection, were excluded from further analysis (12.3% of all trials in monkey T and 5.1% in monkey M). As the duration for which the monkeys were willing to work varied across sessions, after trial exclusion, the analyzed sites included on average 96.4 +/− 48.8 (STD) trials (range 19 to 184) in monkey T, and 147.3 +/− 80.3 trials (range 18 to 281) in monkey M. We also included the sites with few trials, since a majority of the neuronal data analyses were done on trials collapsed across many sites.

Power spectral density (power for short) estimates of the LFP were obtained using the Pwelch function of Matlab. For LFP spectrogram examples (Fig 2A–2C), we highpass filtered the LFP with 3 Hz cutoff, using a fourth order Butterworth filter. Power was estimated for single-trial sliding windows of 300 ms duration, with 50 ms shifts, at 1 Hz resolution, before averaging across trials.

For average spectrograms for each monkey (Fig 2E), we also used 300 ms sliding windows, 50 ms shifts, at 1 Hz resolution. For each individual LFP site, we first highpass filtered the signal (3 Hz cutoff, fourth order Butterworth filter), before calculating the power for each window in single trials. Next, the power matrix (trial × window × frequency) for each LFP was normalized by dividing by the mean power between 10 and 40 Hz across trials and windows for that LFP. We then computed for each window the grand average power across all individual trials for all normalized LFPs (i.e., each trial contributed equally to the grand mean, independent of the total number of trials for the specific LFP site). For single site and average spectrograms, we used a perceptually flat color-map [72], with color limits set to the minimum and maximum power values between 12 and 40 Hz between onset of SEL and GO, separately for each site or each monkey.

To determine the peak frequencies of the 2 observed beta bands, we estimated power in a 900 ms epoch preceding SC1 (blue) onset, across all trials for each LFP site (after highpass filtering at 3 Hz; fourth order Butterworth filter). Within this epoch, we used five 500-ms windows, with overlap of 400 ms, to get 1 average power estimate per trial. We then normalized the power matrix (trial × frequency) for each LFP by dividing by its mean power across trials between 10 and 40 Hz (S3 Fig). To have a "fair" comparison of power across the 2 beta bands, despite the 1/f nature of the aperiodic signal component, the aperiodic signal component was removed in each site independently, before averaging across all trials of all sites and plotting the grand average spectral power (Fig 3A). In order to estimate (and remove) the aperiodic component of the signal, we used an approach similar to FOOOF [73]. Preliminary analysis using the standard FOOOF method showed that it was not estimating adequately the aperiodic component of the spectrum, presumably due to high power in the lower frequencies. Following their assumption that in a restricted frequency range, the aperiodic component is a straight line in the log-log space, and since we were interested in parametrizing the power only for the beta range, we decided to adapt their method to our specific need. We first computed the logarithm of the power of all single trials in each site. Once in the log-log-space, we calculated the

average power per site and looked for 2 local minima; the last minimum before 10 and the last minimum before 50 Hz. The aperiodic component was estimated by a straight line connecting the 2 minima, and then removed from each single trial in that session. In the case one of the 2 local minima were not present in a session, the first line point was set at 10 Hz, and the second in the minimum value between 35 and 50 Hz.

We also determined, for each individual trial, the frequency between 10 and 40 Hz with maximal power (beta peak frequency) in the periodic-only component of the signal (Fig 3A). Based on these distributions of power and peak frequencies, for both monkeys a range for the low band of 13 to 19 Hz and for the high band of 23 to 29 Hz were used to determine the dominant beta band for each LFP site. We computed a beta band dominance index using mean power in the periodic-only signal component across all trials and frequencies in the low band minus mean power across all trials and frequencies in the high band, divided by the sum of the two. Significance in band dominance was determined with a paired $t$ test across trials, taking the mean power across all frequencies in each band for each trial (Fig 3B).

**Phase-locking of neuronal spiking to LFP beta phase.** To verify that the LFP beta bursts were at least partially of local origin, we analyzed phase-locking of the simultaneously recorded neurons to the LFP beta phase of the site-dominant band. We included only the laminar recording sites, and tested phase locking for neurons across all laminar contacts to the LFP on the selected LFP contact on the same laminar probe, to ensure proximity of the 2 signals. We analyzed the delay before SC1 (blue), since the beta amplitude was generally strong in both animals and in both bands in this delay. Only neurons with more than 100 spikes in this delay, accumulated across all trials, were included. Beta phase was extracted from the Hilbert transformation of the beta-filtered LFP, only for the dominant beta band at each LFP site, and the phase at each spike time was determined.

To quantify the phase locking, we first used Rayleigh's test of non-uniformity of circular data [74]. To determine whether the locking was significant for individual neurons, a trial-shuffling method was used. Trial-shuffling is an efficient method for obtaining a "baseline" measure of phase locking, destroying the direct temporal relationship between the 2 signals, while preserving their individual properties such as rhythmicity and dependencies on external (task) events, and 1,000 repetitions of the phase-locking analysis (Rayleigh's test) was done while randomly combining beta phases and spike times from different trials. If the original data yielded a larger z-statistic value from the Rayleigh's test than 950/1,000 (equivalent to $p < 0.05$) of the trial-shuffled controls, the phase-locking of the neuron was considered significant.

**Decoding task condition with beta amplitude.**

*Preprocessing for beta amplitude analysis.* Given the similarity in the behavioral and neuronal data from the 2 animals up to this point, for all subsequent analyses we combined LFPs for both monkeys, while splitting low and high band dominant sites. We furthermore continued the analyses using the single-trial instantaneous beta amplitude. For each LFP site, we first bandpass filtered the signal to extract the dominant beta band, either 16+/− 4 Hz for low dominant sites or 26+/− 5 Hz for high dominant sites, using eighth order Butterworth filters. We next calculated the instantaneous amplitude (envelope) of the beta filtered LFP time series by constructing the analytic signal using the Hilbert transform. The LFP was then cut in trials, before normalizing the beta amplitude by subtracting the grand mean amplitude and dividing by the grand amplitude standard deviation. After normalization, individual trials for all LFP sites with the same beta band dominance were lumped to construct large matrices (trials × time) for each of the 2 beta bands, combining data from the 2 monkeys (Fig 4A).

*Decoding procedure.* First, 2D reduction visualization (t-SNE) was used to explore whether the different color conditions were separable in each of the 2 beta bands (Fig 4B). Second, we

built 2 classifiers using high and low beta bands separately, to decode color conditions. For each, the features were extracted from the temporal evolution of beta amplitude in single trials. We calculated the average beta amplitude in 50 ms non-overlapping time bins from touch to GO in each trial. A random forest estimator from scikit-learn library [75] was trained in the data. Using gridsearch, we found the parameters which maximized classifier performance for both frequency bands (max_depth = 80, max_features = 3, min_samples_leaf = 3, min_samples_split = 8, n_estimators = 200). Unambiguous correct trials (trials in which none of the distractors were presented in the same quadrant) were split in a 60% to 40% ratio between train and test set, respectively. To ensure stability of the method, we repeated the procedure using 20 different data splits, always with class balance in the train set. The average performance for each of the classes was computed by averaging across repetitions. After training the classifier on the unambiguous correct trials, the same model was used to predict distractor error trials (only including the trial in which the 2 distractors were not presented in the same quadrant). In this case, we predicted either the color of the attended (distractor) SC, or the color of SEL; i.e., either the SC the monkey actually used, or the SC the monkey should have used. The chance level was calculated by shuffling the labels in 100 train-test splits of the data for both high and low beta classifiers. All the accuracy values estimated in the different shuffle test-sets were below 0.37, which we set as the overall chance level for the results.

**LM and cross-correlation analysis to quantify the relationship between beta amplitude and behavioral regressors.**

*Dataset preprocessing.* We used the same beta amplitude preprocessing of individual trials described in the previous section for the next analysis. Moreover, the eye signals (position and velocity) were upsampled to 1 KHz, to have the same temporal resolution as the LFP and hand signals. The eye velocity was upsampled using a linear interpolation, whereas the position of the gaze in the different clusters of the work area was upsampled using the nearest neighbor interpolation.

*Bayesian index criterion.* To evaluate the relation between complementary continuous and categorical variables with the LFP signal, we performed an LM analysis. The LFP from either low high beta bands were the variables to explain. The regressors considered to explain the data were 7: color conditions (3 levels), movement direction (correct target location, 4 levels), reaction time (normalized with a z-score inside each recording session), time-on-task (computed with the relative position of the trial within the recording session), hand velocity (cm/s), eye velocity (cm/s), and the gaze position of the animal (inside versus outside the work area, 2 levels) and all 2-by-2 interactions. More complex interactions were excluded from the model to simplify the interpretation of the results and reduce the number of potential regressors. We considered a total of 6,800 ms, from −1,200 ms to 5,600 ms from the SEL for the analysis. All the neuronal and behavioral data were then binned in 10 ms non-overlapping windows. In each bin, we applied a total of 247 models (all combinations of 1 to 7 regressors including or not their 2-by-2 interactions) and compared them using a BIC. The BIC is sensitive to the number of trials considered in each model fitting. Consequently, we applied the same selection for each model, removing from all bins the trials in which the eye or the hand signals were missing, and furthermore removing trials in individual bins if the eye signal was saturated because of an eyeblink or an extreme eye position outside the dynamic range of the eye camera. We then examined the presence or not of a regressor and or interactions in the winning model in each of the 680 bins. This first analysis allowed us to target the regressors explaining the most trial-by-trial variability of high and low beta amplitude (Fig 4B).

*Linear model analysis.* Based on the BIC analysis, 5 regressors, without interactions, were selected. Movement direction, eye velocity, and all the possible pairwise interactions were discarded because they were rarely represented in the regressors best explaining the beta. The

trial selection was different for each selected regressor, based on available trials for each regressor. All trials could be used for color condition, RT, and time-on-task (trial number within each session) regressors. Good quality of the hand signal was necessary for the hand velocity regressor. Good quality of the eye signal was necessary for the gaze position. For gaze position, the eyeblinks were considered as outliers and the corresponding single-trial bins with an eyeblink were removed for the model fitting. The different number of trials available considered for each bin and each regressor prompted us to consider each regressor separately. For each bin, each regressor and each beta band, we applied a regression model (*fitlm*) to describe the relationship between beta amplitude and the 5 different predictors. The color condition was applied considering all conditions together. The other models were applied considering each color condition separately. Considering that some variables were categorical, we applied an ANOVA to the model objects to test the significance of the categorical variables. *P*-values <0.01 were considered significant.

*Two-dimensional cross correlograms between hand velocity and low beta amplitude.* An analysis equivalent to the joint peristimulus time histogram (eq-jpsth) representation of the cross-correlation between 2 neurons [76,77] was performed using the Fieldtrip toolbox [78]. Instead of the activity of 2 simultaneously recorded neurons, we used hand velocity and low beta amplitude as input signals (or simultaneously recorded high and low beta in S5 Fig). The eq-jpsth corresponds to the 2D cross-correlograms. Time versus lag are represented on the abscissa and ordinate and the color axis represents the strength of the correlation. The *raw* eq-jpsth was first computed and then, the trials were shuffled for 1 variable and the same matrix was obtained, the *shuffled* eq-jpsth. This procedure was performed 100 times to obtain a distribution of 100 *shuffled* matrices. The corrected matrix was obtained by subtracting the mean of the 100 *shuffled* matrices from the *raw* matrix and to divide this *subtracted* matrix by the square root of the cross product of the time-dependent variance of the *raw* matrix. The scale of the *subtracted* eq-jpsth is thereby bounded between −1 and 1 and named correlation coefficients. At each point in the *subtracted* eq-jpsth, a correlation was considered significant if the value in that point in the *raw* matrix (before correction) was always superior or always inferior to the 100 values from the *shuffle* matrices in the same point (Fig 6C–6E). The data along the diagonal and paradiagonal of the *subtracted* matrix were averaged to obtain cross correlograms (Fig 6F–6H). The lag with the largest negative value (anticorrelation) was determined in the trial period prior to and after valid SC onset.

## Supporting information

**S1 Fig. Additional behavioral results. Related to Fig 1.** (A) Average hand velocity in 1 example session in monkey T, split for the 3 color conditions. On the left, zoomed in to the micro-movements performed during the trial between central touch and GO. To the right with velocity scale adjusted to the final center-out reaching, aligned to movement onset. (B) Hand velocity in a randomly selected subset of correct green trials in the same session as in A. The 2 solid black vertical lines connected with a horizontal arrow reflect the epoch used to estimate X and Y offset (drift) of micromovements in the post-cue epoch (in D). The velocity scale is indicated inside the plot. (C) Average hand velocity in the 1-s delay after each SC, split for color condition, averaged across all trials for all behavioral sessions for the 2 monkeys combined. Horizontal black lines on top of the bar plots denote significant differences in single-trial hand velocity. (D) Hand cursor displacement (drift) caused by micro-movements across all green trials in each monkey, split according to the target direction. Each dot reflects 1 trial, and the position reflects the relative X and Y offset 1 s after the onset of SC2 (second vertical solid black line in B), compared to the position at SC2 onset (first vertical solid black line in B). UR,

upper right; LR, lower right; LL, lower left; UL, upper left. The total number of trials is indicated (*n*). (E) Average deltoid EMG amplitude, recorded in the same behavioral session as shown in A and B, on the left for the period between touch and GO (split for the 3 color conditions) and on the right aligned to movement onset (averaged for the 3 color conditions). Towards the body (LL) in solid lines and away from the body (UL) in dotted lines. The raw EMG signal (30 kHz) was first rectified, and then low-pass filtered at 250 Hz and downsampled to 1 kHz. A Gaussian filter (length 150 ms, width 100 ms) was used to smooth single trials before plotting the trial-averaged EMG. Source data are available in S2 Data.
(TIF)

**S2 Fig. Condition decoding by uninstructed behaviors in correct trials. Related to Fig 1.** (A) Left. Average hand velocity across all trials in all behavioral sessions. Data from both monkeys are combined (*n* = 11,587). Gray rectangle represents the epoch considered for the following decoding analysis. Right. Decoding performance of SEL (color condition) category in correct trials using the temporal profile of hand velocity. Performance is presented as proportions of the total number of trials of each category in the test set (totaling 1 for each row). The diagonal represents the true positive accuracy, and the off-diagonal values correspond to the proportions of trials of each category incorrectly assigned to another category. The estimated chance level (0.35) is marked on the color scale bar. (B) Left. Gaze position for both monkeys combined across all trials in all behavioral sessions with eye movement recordings (*n* = 8,630). Each curve represents the proportion of trials in which the gaze position was inside the working area (number of trials In/number of trials In + Out) along the trial. Eyeblinks were considered as missing data. Right. Decoding performance of SEL (color condition) category in correct trials using the temporal profile of gaze position. Other parameters are the same as in A right. Source data are available in S2 Data.
(TIF)

**S3 Fig. Average normalized power and peak frequency distribution of the full signal. Related to Fig 3.** (A) Average normalized power in the pre-SC1 period across all trials for all sites in each monkey, for the full LFP signal, including aperiodic and periodic components. The curves reflect the mean power ±SEM across LFP sites. Overlain are distributions of single-trial peak frequency (frequency with maximal power) between 10 and 40 Hz in the same task period for the full LFP signal. Source data are available in S2 Data.
(TIF)

**S4 Fig. Average low and high beta amplitude, and main regressors explaining amplitude variance. Related to Figs 4 and 6.** (A, C) Same as Fig 4A, separated for Monkey T (A) and Monkey M (C). (B, D) Same as Fig 6B, separated for Monkey T (C) and Monkey M (D). (A, C) Representation of the trial-averaged temporal profile of normalized high (left; 21–29 Hz) and low (right; 12–20 Hz) beta amplitude (+/− SEM), separated by the 3 color conditions. The horizontal gray line above each plot graph represents the time-resolved modulation in beta amplitude by the color condition along the task. The significativity is represented as in Fig 4A. (B, D) Time-resolved representation of the presence of each regressor in the winning model after the application of a Bayesian index criterion (BIC) for the comparison of all possible models and their 2-by-2 interactions, for the high beta (left) and the low beta (right). Each row represents a regressor, the last row represents all possible interactions. Regressors selected for ulterior analysis in orange and discarded regressors in gray. Source data are available in S2 Data.
(TIF)

**S5 Fig. Correlation between simultaneously recorded high and low beta. Related to Fig 4.** Equivalent of a joint peristimulus time histogram (jpsth) applied to the high and low beta

amplitude. The analysis was performed separately for both monkeys (A–C: Monkey T; D–F: Monkey M) and the 3 colors (A, D: blue; B, E: green; C,F: pink). Each point of the matrix represents the corrected trial-by-trial cross product of the 2 variables on 50 ms bins. Each colored matrix point was inferior (cold color) or superior (warm color) to all 100 values from shuffled matrices (equivalent $p$-value of 0.01). The vertical and horizontal lines represent the appearance and disappearance of the valid SC for the 3 conditions. Vertical color bars on the right of each jpsth represent the significativity of the correlation between low beta and hand velocity, for each color condition. The significativity is represented as a color-scale gradient (brightest color for $p = 0.01$ and darkest color for $p < = 1e-08$; white means nonsignificant). Source data are available in S2 Data.
(TIF)

**S6 Fig. Average MUA in high and low beta dominant sites. Related to Fig 4.** Average MUA amplitude including all trials of all recording sites, for high beta band (left) and low band (right) dominant sites, separately for the 3 color conditions. The MUA was generated following the method of Stark and Abeles (2007) [79]. The raw signal was first bandpass filtered (300–6,000 Hz) and clipped beyond +/− 2 standard deviations. Then, the signal was squared, smoothed with a low-pass filter (250 Hz) and downsampled from 30 to 1 kHz, before the square-root was taken to arrive at the final MUA signal. The MUA from each site was cut in trials, and normalized by dividing by the mean amplitude across all trials and trial-times. Finally, the single-trial MUA from all sites with the same LFP beta band dominance were combined. The plots show trial-averaged MUA after first smoothing individual trials with a Gaussian filter (length 30 ms, width 15 ms). Source data are available in S2 Data.
(TIF)

**S7 Fig. Variance explained by full model with each regressor scrambled across trials. Related to Fig 6.** (A) Total percentage of variance explained by the 4 selected regressors, RT, hand Velocity, gaze position, and time-on-task, along the task, in the 3 different color conditions for both bands. (B–E) Percentage of variance explained by the full model minus the full model in which the values of one regressor were scrambled across trials. For each bin, the values were scrambled 100 times and the average of the 100 scrambles was subtracted to values obtained with the full model. The dots above each graph represent an equivalent $p$-value of 0.01. For a temporal bin, if the value of variance explained obtained with the full model was superior to all the 100 values obtained with the regressor of interest scrambled, the effect of the regressor in that bin was considered significant and marked with a dot. From B to E, the values were scrambled respectively for the RT, hand velocity, gaze position, and time-on-task. Source data are available in S2 Data.
(TIF)

**S8 Fig. Influence of RT on low beta depending on the presence of other regressors. Related to Fig 7.** Comparison of the significativity of RT as the unique regressor of a linear model vs. paired with each of the other regressors, separated by color conditions. In each plot, the $p$-value is displayed on a logarithmic scale, in color when RT was the unique regressor considered and in black when paired with a second regressor. The horizontal dotted lines on the top of each plot represent the significance of each case ($p < 0.01$). Red horizontal dotted lines are plotted for $p$-values equal to 1, 0.05 and 0.01. (A) RT paired with hand velocity as a second regressor. (B) RT paired with the gaze position as second regressor. (C) RT paired with time-on-task as second regressor. Source data are available in S2 Data.
(TIF)

**S9 Fig. Correlations between beta amplitude and gaze position. Related to Fig 8.** (A) Representation of the negative, zero, and positive lag meaning in the correspondence between LFP (top) and gaze position (bottom). Three trials are represented for beta and gaze position. A negative lag means relating the gaze position with LFP in the past (before), beta was consequently leading gaze. Zero lag means relating LFP and gaze position from the same temporal bin. A positive lag means relating gaze position to the LFP in the future (later), gaze was consequently leading beta. (B) Proportion of bins in which beta amplitude (both bands combined) modulated significantly with gaze position, for different temporal lags. Beta was leading gaze for negative values (i.e., gaze at time t0 and LFP at time t0—lag, represented in orange). Correlation at zero lag is represented in gray. Gaze was leading beta for positive values (i.e., gaze at time t0 and LFP at time t0 + lag, represented in red). (C) Low beta band split into groups of trials based on the position of the gaze at 200 ms after the onset of the valid SC. Brown curves represent the trials in which the monkeys were looking inside the working area (either on the target or elsewhere). Purple represents the trials in which the monkeys were looking outside the working area. (D) High beta amplitude split into groups based on the monkey's position of the gaze at different time lags (from top to bottom; −1,000 ms, −240 ms, 0 ms, 240 ms, 1,000), all color conditions combined. Red dashed vertical lines represent the values from and up to which significant bins have been counted in B. We excluded the first and last 1,000 ms of the period spanning from −1,200 ms to 5,600 ms from SEL. (E) Same representation for the low beta band. Source data are available in S2 Data.
(TIF)

**S10 Fig. Spectral parametrization for early and late trials. Related to Fig 9.** (A) Spectral parametrization using the FOOOF method [73] for the pre-SC1 period of blue trials. The analysis was done separately for the first third of trials for each session (early; left) and the last third of trials for each session late; right), for each monkey separately. The black line corresponds to the original data and the red line to the model fit. The algorithm identifies the aperiodic signal (blue dashed line) and the spectral peaks and their peak frequency (green). A frequency range of 5–194 Hz was used for fitting the data, using the "knee" mode. (B) Spectrum decomposition in periodic (left) and aperiodic (right) signal components, in early and late blue trials in the sessions, for each monkey. The frequency axis was cut at 45 Hz for the periodic signal to focus on the lower frequencies including the beta bands. Source data are available in S2 Data.
(TIF)

**S1 Table. Behavioral task performance. Related to Fig 1.** Summary of all errors, number of correct trials included for behavioral analyses, percent of distractor errors and RTs (+/− SD) for each color condition and movement direction for each animal. UR, upper right; LR, lower right; LL, lower left; UL, upper left.
(DOCX)

**S2 Table. Reference position for each LFP site. Related to Methods.** Summary of recording reference positions for low and high beta dominant sites in each monkey, further separated for linear array probe and single-tip electrode sites. Tube—stainless steel body of Plexon probe; chamber—on screw of the titanium recording chamber, or in contact with chamber saline; headpost—skull screw of headpost.
(DOCX)

**S1 Data. Data for the reproduction of all main figures.**
(XLSX)

**S2 Data. Data for the reproduction of all supplementary figures.**
(XLSX)

**S1 Script. Custom code to compute the aperiodic estimation.**
(PY)

## Acknowledgments

The authors thank Joel Baurberg, Xavier Degiovanni, and Luc Renaud for technical assistance; Sébastien Barniaud, Laurence Boes, Frédéric Charlin, and Marc Martin for animal care.

## Author Contributions

**Conceptualization:** Simon Nougaret, Bjørg Elisabeth Kilavik.

**Data curation:** Bjørg Elisabeth Kilavik.

**Formal analysis:** Simon Nougaret, Laura López-Galdo, Emile Caytan, Bjørg Elisabeth Kilavik.

**Funding acquisition:** Bjørg Elisabeth Kilavik.

**Investigation:** Julien Poitreau, Bjørg Elisabeth Kilavik.

**Software:** Simon Nougaret, Laura López-Galdo, Frédéric V. Barthélemy, Bjørg Elisabeth Kilavik.

**Supervision:** Frédéric V. Barthélemy, Bjørg Elisabeth Kilavik.

**Writing – original draft:** Simon Nougaret, Bjørg Elisabeth Kilavik.

**Writing – review & editing:** Laura López-Galdo, Emile Caytan, Julien Poitreau, Frédéric V. Barthélemy.

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
