## [Editor Report · Decision Letter 0]

19 Oct 2023

Dear Dr Kilavik, 

Thank you for submitting your manuscript entitled "Distinct sources and behavioral correlates of

 macaque motor cortical low and high beta" for consideration as a Research Article by PLOS Biology.

Your manuscript has now been evaluated by the PLOS Biology editorial staff as well as by an academic editor with relevant expertise and I am writing to let you know that we would like to send your submission out for external peer review.

Once your full submission is complete, your paper will undergo a series of checks in preparation for peer review. After your manuscript has passed the checks it will be sent out for review. To provide the metadata for your submission, please Login to Editorial Manager (https://www.editorialmanager.com/pbiology) within two working days, i.e. by Oct 21 2023 11:59PM.

Kind regards,

Christian

Christian Schnell, PhD

Senior Editor

PLOS Biology

cschnell@plos.org

---

## [Decision Letter · Decision Letter 1]

11 Dec 2023

Dear Dr Kilavik,

Thank you for your patience while your manuscript "Distinct sources and behavioral correlates of macaque motor cortical low and high beta" was peer-reviewed at PLOS Biology. It has now been evaluated by the PLOS Biology editors, an Academic Editor with relevant expertise, and by several independent reviewers. 

In light of the reviews, which you will find at the end of this email, we would like to invite you to revise the work to thoroughly address the reviewers' reports.

As you will see below, the reviewers find your study interesting and important but raise a number of important concerns that need to be addressed.

Given the extent of revision needed, we cannot make a decision about publication until we have seen the revised manuscript and your response to the reviewers' comments. Your revised manuscript is likely to be sent for further evaluation by all or a subset of the reviewers.

**IMPORTANT - SUBMITTING YOUR REVISION**

*Re-submission Checklist*

*Published Peer Review*

*PLOS Data Policy*

*Blot and Gel Data Policy*

Sincerely,

Christian

Christian Schnell, PhD

Senior Editor

PLOS Biology

cschnell@plos.org

REVIEWS:

Reviewer #1: This manuscript is of potential interest to the field of motor control in primates (including humans). As introduced in the manuscript, there has been a long-standing interest in the mu/alpha/beta band; however, there is confusion about the roles of the low vs higher frequency. This manuscript suggests that there may be differences in the premotor vs the motor regions. In general, this finding may be of interest to multiple fields. However, there are some fundamental methods details and analyses that needs clarification. Without these details, it is hard to fully understand the significance. 

1. As the authors are aware, low and high beta could be simply be harmonics (consequence of fourier transform of a signal). For example, a strong mu (low beta) could have a waveform shape that also result in a high beta band harmonic. In other words, the closer a waveform deviates form a pure sine wave, the more there are harmomics at multiples. This might be suggested by Figure 2 and Figure 3A. Examining this in detail and showing waveforms is key. What are the LFP waveforms in each area? For PMd, is there really no fundamental mu band and only beta?

2. The beta band index is particularly sensitive to point #1. How does waveform shape relate to this index.

3. I was confused by the methods and whether there was referencing or not. Given all the types of electrodes used, this is important. Please make a table with each animal/recording approach/note of GND and referencing or not. This can have important effects on the LFP signal � then will affect filtering.

4. Using phase for local effects might also be problematic because of the harmonic and the phase consistency b/w high and low beta. The authors should examine cross-frequency (bi-spectra) between these two bands per channel. Is there a consistent phase b/w the two?

5. Was there any food or water restriction or positive rewards? This was not clear from the methods.

6. Can the authors show example plots of the movement threshold. The values in Figure 1 imply velocities of 0.2 and 0.6 cm/s. 

7. For some reason, I could not find figure legends. Thus, not clear what all the elements of the figure mean. Might be easiest if they are part each figure.

Reviewer #2: This is a very nice paper, highlighting the distinction between low and high beta more clearly than any other paper I know. I have mainly interest-driven comments - the paper actually looks pretty good to me as it is. 

Comments:

I assume that the animals' choice could be equally well predicted from pre-GO velocity, i.e. the onset of micro-movements? This would emphasize the influence of motor processes on choice decoding, which I feel is sometimes underrated. 

I wonder if the authors found any gamma responses/oscillations and, if so, whether these gamma oscillations behaved like a negative image of high-beta. 

In Fig. 5, left column (high beta), some of largest differences between conditions arise before the SEL cue. At that time, the animal had no information about the kind of trial, right? So why do the brain oscillations distinguish between conditions at this stage?

It would be nice to see a grand-average power spectrum across high beta dominant and low beta dominant sites to appreciate how strong the separation really is. 

In Fig S5, top row, the authors display a "full" model, including all selected predictors, to better understand the hierarchy of predictors. I assume it might be helpful to also see the influence of each predictor (coefficients) vs. time, in addition to the variance explained. 

Minor:

"For the high band dominant sites, 47.7% of neurons (27/60 neurons in monkey T and 45/91 in monkey M) were significantly phase-locked to high beta phase. For the low band dominant sites, 12.3% of neurons (22/269 neurons in monkey T and 38/218 in monkey M) were significantly phase-locked to low beta phase." The number of units locked to the respective "non-dominant" rhythm should also be reported. 

It seems that the referencing was quite variable across recordings. Why was it changed so often?

p. 32: "Next, the power matrix (trial x window x frequency) for each LFP was normalized by dividing by the mean power between 10-40Hz across trials and windows for that LFP."

Was the same value used to normalize power at all frequencies of the time-frequency spectrum? Usually, one would prefer a normalization per frequency. 

p. 25, bracket missing: "Continuous hand position (X and Y) was obtained from two perpendicularly superimposed contactless linear position magnetorestrictive transducers,model MK4 A; GEFRAN, Provaglio d'Iseo, Italy)."

Reviewer #3: Review for 'Distinct sources and behavioural correlates of macaque motor cortical low and high beta' by Nougaret and colleagues for PLOS Biology

Summary

The authors investigate beta activity over 59 and 39 sessions using multi-electrode recordings from two adult male rhesus monkeys performing a complex visuomotor task.

The authors report distinct beta activity:

1. low beta located in M1 which correlates positively with reaction time and negatively with micro-movements

2. high beta located in the dorsal premotor cortex which relates to task prediction.

First, I want to apologize to the authors and to the editor for the slight delay - I needed to

sleep on this one a bit. I also would like to point out that I'm not an expert in the acquisition and analysis of eye position data.

I think this is excellent work in an interesting line of questioning. The paper is generally well-written, and the visual illustrations are exceptional.

However, despite reading the results section several times, I find it quite hard to grasp all the information and make sense of why specific analyses were conducted.

One aspect that contributed to this is that the results are presented prior to the methods. I understand that this is journal-dependent rather than the author's choice. However, I would highly recommend adding enough information in a short and compact way to the respective results section. This will increase the flow and prevent the reader must jumping back and forth between the results and methods sections.

Further, I wonder if some results could be moved to the supplementary material.

Finally, I strongly feel that each analysis should be well-motivated and summarised. Adding systematically two sentences: one sentence before an analysis explaining why the following analysis was conducted; and one sentence after an analysis summarising what this analysis yield would be extremely helpful. At times the results section reads like a collection of analyses that are presented back-to-back, whereby the reader is sometimes wondering why a particular analysis has been conducted.

I am a bit perplexed by the very different time courses of low and high beta (for instance as presented in Fig. 4A). If I understood correctly the authors present data from different frequencies (low, high) at different locations (M1, PMd). However, there is no figure showing the low beta in PMd and the high beta in M1. I think it would be useful to show these data, as well as statistically compare the time courses of low and high beta in the same spatial location. This allows unpicking if the reported differences between low and high beta arise from different spectral properties of the signal, the different spatial properties of the signal, or both. To me, this is a very important point the paper should incorporate given the title and claims of the paper.

Check the spelling of occured > occurred

---

## [Editor Report · Decision Letter 2]

2 Apr 2024

Dear Bjørg,

Thank you for your patience while we considered your revised manuscript "Distinct sources and behavioral correlates of macaque motor cortical low and high beta" for publication as a Research Article at PLOS Biology. This revised version of your manuscript has been evaluated by the PLOS Biology editors and the Academic Editor.

Based on our Academic Editor's assessment of your revision, we are likely to accept this manuscript for publication, provided you satisfactorily address the following data and other policy-related requests.

* We would like to suggest a different title to improve readability: "High and low beta frequency oscillations in the cortex have specific roles in both movement control and spatio-temporal attention"

* Please include the full name of the IACUC/ethics committee that reviewed and approved the animal care and use protocol/permit/project license. Please also include an approval number.

DATA POLICY:

Thank you for making the data and scripts available at https://gin.g-node.org/In2PrimateBrains/. Could you please add some description and specification where to find the required files? It is currently not clear which of the files belong to your study.

CODE POLICY

Per journal policy, if you have generated any custom code during the curse of this investigation, please make it available without restrictions upon publication. Please ensure that the code is sufficiently well documented and reusable, and that your Data Statement in the Editorial Manager submission system accurately describes where your code can be found. 

We expect to receive your revised manuscript within two weeks. 

*Published Peer Review History*

*Press*

Sincerely,

Christian

Christian Schnell, PhD

Senior Editor

cschnell@plos.org

PLOS Biology

---

## [Editor Report · Decision Letter 3]

8 May 2024

Dear Dr Kilavik,

Thank you for the submission of your revised Research Article "Low and high beta rhythms have different motor cortical sources and distinct roles in movement control and spatio-temporal attention." for publication in PLOS Biology. On behalf of my colleagues and the Academic Editor, Emmanuel Procyk, I am pleased to say that we can in principle accept your manuscript for publication, provided you address any remaining formatting and reporting issues. These will be detailed in an email you should receive within 2-3 business days from our colleagues in the journal operations team; no action is required from you until then. Please note that we will not be able to formally accept your manuscript and schedule it for publication until you have completed any requested changes.

Please do not forget to assign the DOI to your repository and replace the links in the manuscript accordingly.

PRESS

Kind regards,

Christian

Christian Schnell, PhD

Senior Editor

PLOS Biology

cschnell@plos.org